# Antiparallel stacking of Csu pili drives *Acinetobacter baumannii* 3D biofilm assembly

Henri Malmi [1], Natalia Pakharukova [1,2,8], Bindusmita Paul [3,4,8], Minna Tuittila[1], Irfan Ahmad[5,7], Stefan David Knight [6], Bernt Eric Uhlin [5], Debnath Ghosal [3,4] & Anton V. Zavialov [1] ✉

Many Gram-negative nosocomial pathogens rely on adhesive filaments, known as archaic chaperone-usher pili, to establish stress- and drug-resistant, multi-layered biofilms. Here, we uncover the mechanism by which these pili build three-dimensional (3D) biofilm architectures. In situ analyses of *Acinetobacter baumannii* biofilms using electron microscopy (EM) reveal an extensive network of ultrathin, flat stacks of archaic Csu pili interconnecting bacterial cells in 3D space. Cryo-EM structures of a single native pilus, pilus pairs, and two types of multi-pilus stacks show that the pili pack into antiparallel sheets, with their rods connected laterally by junctions at their zigzag corners. This antiparallel arrangement ensures that contacts form primarily between pili from interacting cells rather than pili from the same cell. With a remarkably short helical repeat, archaic chaperone-usher pili spontaneously establish a high density of junctions that determines the biofilm's 3D architecture. Our findings may help develop new therapies against multidrug-resistant bacterial infections by targeting pilus-pilus interactions.

The formation of multi-layered biofilms is an effective survival strategy for many bacterial pathogens[1–3]. Dense 3D biofilms shelter bacteria[4] from anti-bacterial drug treatments and harsh environments[5], while promoting horizontal gene transfer[6] and development of resistance mutations through exposure to lowered drug doses[7], often leading to poor bacterial eradication and chronic infections[8,9]. A variety of extracellular polymeric substances (EPS), including filamentous proteins, polysaccharides, and extracellular DNA (eDNA), have been found to play a role in 3D biofilm formation in different bacteria[10]. However, the biofilm matrix composed of these EPS is not easily amenable to

high-resolution structural analysis, and therefore little is known about the mechanisms of bacterial cell-cell association and 3D biofilm architecture[10].

Adhesive pili, or fimbriae, are hair-like surface appendages that mediate bacterial spread and colonisation of host tissues. In Gram-negative bacteria, most adhesive pili assemble from protein subunits via the chaperone-usher pathway (CUP)[11–13]. CUPs are subdivided into three groups comprising six major phylogenetic clades: alternative (α-fimbriae), classical (β-, γ-, κ-, and π-fimbriae), and archaic (σ-fimbriae)[12]. While classical and alternative CUPs are primarily found in β- and γ-

---

[1]Joint Biotechnology Laboratory, MediCity, Faculty of Medicine, University of Turku; Tykistökatu 6A, Turku, Finland. [2]Department of Medicine, Duke University Medical Center, Durham, North Carolina, USA; Howard Hughes Medical Institute, Duke University Medical Center, Durham, North Carolina, USA. [3]Department of Biochemistry and Pharmacology, Bio21 Molecular Science and Biotechnology Institute, University of Melbourne, Parkville, Melbourne, Australia. [4]Australian Research Council (ARC) Centre for Cryo-Electron Microscopy of Membrane Proteins, Bio21 Molecular Science and Biotechnology Institute, University of Melbourne, Parkville, Melbourne, Australia. [5]Department of Molecular Biology and Umeå Centre for Microbial Research (UCMR), Umeå University, Umeå, Sweden. [6]Department of Cell and Molecular Biology, Biomedical Centre, Uppsala University; Husargatan 3, Uppsala, Sweden. [7]Present address: Department of Cell and Molecular Biology, Science for Life Laboratory, Uppsala University, Uppsala, Sweden. [8]These authors contributed equally: Natalia Pakharukova, Bindusmita Paul. ✉e-mail: anton.zavialov@utu.fi

proteobacteria, archaic CUPs are more prevalent and present across a wider range of phyla[12].

Classical and alternative CUPs primarily facilitate initial, highly specific adhesion. In contrast, archaic CUP systems are associated with the establishment of mature 3D biofilms[14–17]. Among the most troublesome Gram-negative bacteria, *Acinetobacter baumannii* and *Pseudomonas aeruginosa*[18,19] rely on archaic Csu and CupE pili, respectively, to develop stress and drug-resistant mushroom or mountain-like 3D biofilms on a diverse array of biotic and abiotic surfaces[14–17,20]. Although the central role of Csu and CupE pili in 3D biofilm formation has been firmly established through deleterious mutagenesis of structural or assembly genes from both archaic systems[14–16], the underlying mechanism of this process remained elusive.

The initial surface attachment of *A. baumannii* is mediated by hydrophobic finger-like loops at the tip of the two-domain subunit CsuE[21], which is displayed by adaptor subunits CsuA and CsuB at the tip of a long Csu pilus rod made of subunit CsuA/B[22]. Both Csu and CupE pili are assembled through the archaic CUP[20,21,23,24] by donor strand complementation (DSC), in which the incoming pilin subunit inserts its donor strand into the hydrophobic groove of the preceding subunit, thereby completing its immunoglobulin-like fold[25–29]. Unlike the classical and alternative CUP pili that have either tube-like helical[30–32] or flexible rods[33], the Csu and CupE pili share a characteristic zigzag architecture with short 7-subunit and 5-subunit repeats, respectively, allowing them to act as super-elastic springs under tensile forces[22,34].

Here, we elucidate the mechanism of biofilm maturation into multi-layered structures using a combination of scanning, negative stain and cryogenic electron microscopy of *A. baumannii* biofilms and isolated networks of Csu pili. We reveal that Csu pilus rods organise into flat antiparallel sheets linking neighbouring cells in three-dimensional space. Meanwhile, poly-N-acetylglucosamine (PNAG) fills the gaps within the Csu stack networks, collectively providing protection to the bacteria.

## Results

### Flat stacks of Csu pili interconnect *A. baumannii* cells in biofilms

Earlier studies of *A. baumannii* biofilms using scanning electron microscopy (SEM) have shown that bacterial cells are interconnected by Csu pili[20,35]. However, our analysis of SEM images indicates that the thickness of these 'pili' is often much greater than the width of a single Csu pilus rod filament determined by cryo-EM[22]. In the example shown in Fig. 1, the apparent width is ~30 nm, which is ten-fold greater than that of a single pilus. This prompted us to examine the Csu pili on bacteria grown in liquid and on solid media at higher resolution using negative staining transmission electron microscopy (TEM) (Fig. 2).

*A. baumannii* 19096 T translucent phenotype cells grown in a liquid medium display Csu pili that are mostly separate and pointed radially in every direction (Fig. 2a). In contrast, in colonies of these cells, most Csu pili stack upon each other, merging into flat sheets

(Fig. 2c–e, Supplementary Fig. 1a–c). This generates a network of stacks in the tight gaps of a bacterial colony, with relatively thin, local stacks that connect bacterial neighbours (Fig. 2g, i) merging into long and wide "superstacks" (Fig. 2f, h, Supplementary Fig. 1b, c) that interconnect multiple bacteria (Fig. 2b, Supplementary Fig. 1d) and essentially wrap around each bacterium, akin to a sheath (Fig. 2c, Supplementary Fig. 1d–h). Stacks may contain more than 15 pili, with pili often forming lateral interactions extending over half of their length (Supplementary Fig. 1a–c). Superstacks, continually incorporating new pili from the surfaces of multiple bacteria, can exceed 12 μm in length (Supplementary Fig. 1d–h), clumping bacteria in 3D structures (Supplementary Fig. 2). These structures formed equally efficiently under both ambient and physiological temperatures (22–37 °C) and across a broad range of pH (6.0–8.5) and salt concentration (42.8–400 mM) (Supplementary Fig. 2).

To study the Csu pilus stacks in their native state, we analysed the suspended *A. baumannii* colonies using cryo-electron tomography (cryo-ET) (Fig. 3). 3D segmentation of reconstructed tomograms revealed individual pili stacking into flat, single-layer sheets (Fig. 3d–i, Supplementary Videos 1, 2). These sheets widen as they extend from the bacterial surface and merge into larger sheets, interconnecting multiple bacteria. Thus, altogether, our results suggest that Csu pili connect bacterial cells in *A. baumannii* biofilms through a complex 3D network of flat stacks.

### Csu pilus–mediated bacterial interlinking and adhesion to surfaces depend on different structural elements of the pilus

While both piliated *A. baumannii* and *csu*-expressing *Escherichia coli* typically exist as single cells in liquid cultures, they occasionally form small clumps, resembling the initial stages of the formation of a surface pellicle[36]. To investigate the nature of these clumps, we analysed them using negative staining TEM (Supplementary Fig. 3). The cells within these clumps had interconnecting Csu pilus stacks with a similar architecture to the stacks seen in 3D biofilms. This suggests that prior adhesion to surfaces is not strictly necessary for Csu pili to link the bacteria together.

This observation was further supported by the detection of similar Csu pilus stack-tethered bacterial clumps (or rafts) in liquid cultures of *E. coli* strain expressing recombinant Csu pili rendered unable to facilitate bacterial adhesion to hydrophobic surfaces due to a mutant tip-subunit CsuE with a compromised hydrophobic tip-finger site[21] (Supplementary Fig. 4a–d). Similarly, extensive Csu pilus stacks formed in the induced colonies of the recombinant *E. coli* strain, regardless of whether it possessed the wild type or the mutant CsuE (Supplementary Fig. 4e, f). Collectively, these results indicate that Csu pilus–mediated surface adhesion and bacterial interlinking are distinct processes that depend on different structural elements of the pilus. Nevertheless, both processes are essential for 3D biofilm formation.

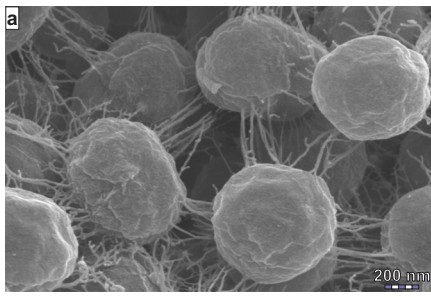
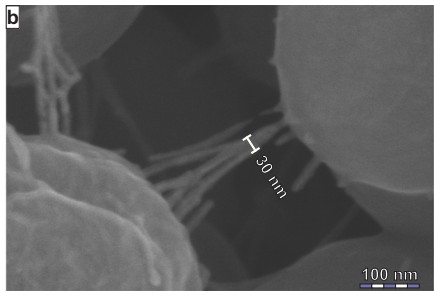

**Fig. 1 | Scanning electron microscopy analysis of *A. baumannii* 3D biofilms. a**, **b** SEM images of *A. baumannii* clinical isolate A100 showing (**a**) Csu pilus-pilus interactions between bacterial cells in the biofilm, and (**b**) the stacked pili interactions at higher magnification. The bacteria were grown at 30 °C for 72 h before

samples were fixed and prepared for SEM analysis as described in "Methods". Representative SEM images were selected from micrographs of grids prepared from three individual colonies.

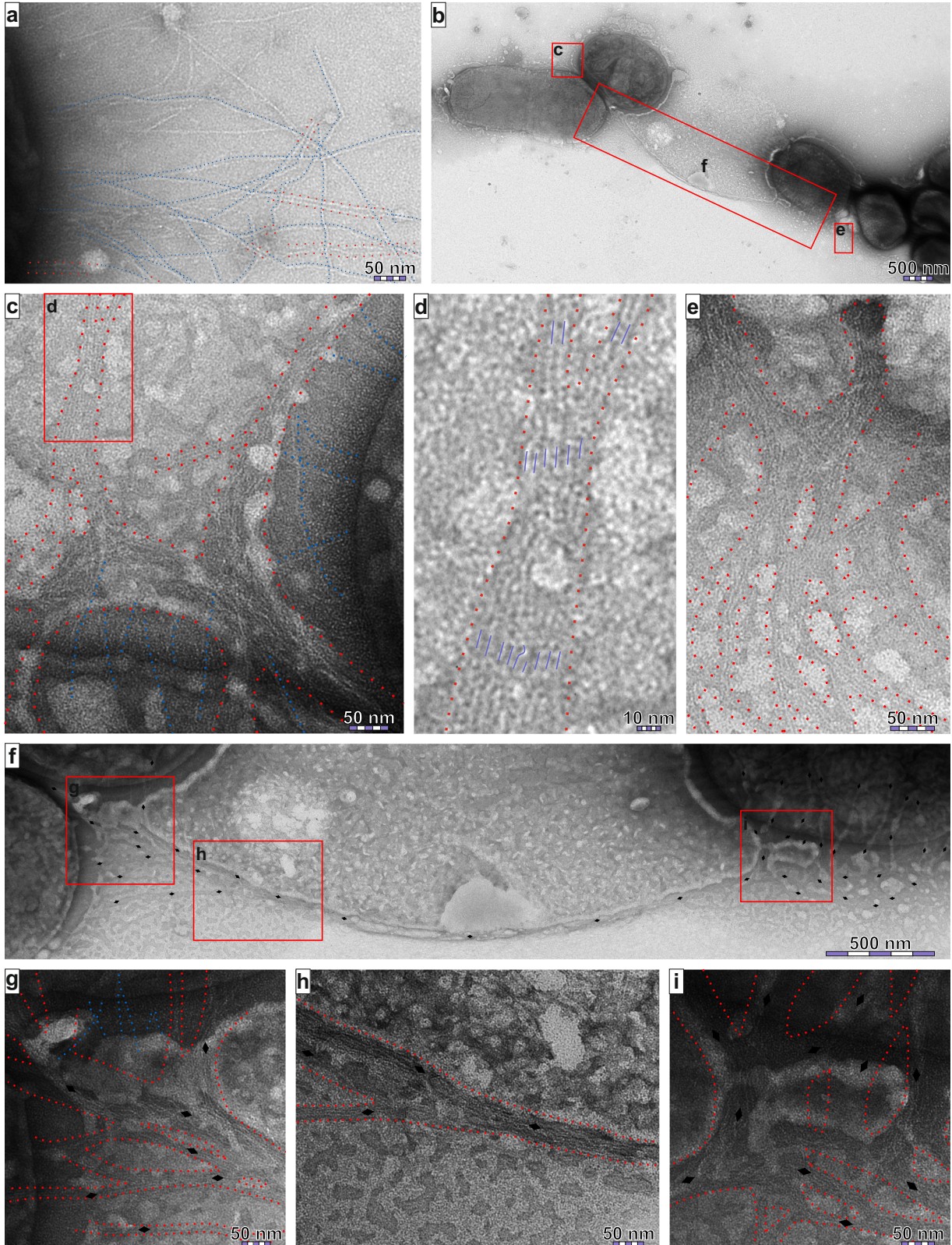

## Csu pili spontaneously form a network of pilus stacks in vitro

To study the mechanism of Csu pilus stack formation, we isolated Csu pili from *A. baumannii* and recombinant *E. coli* cells and examined these samples at different time points using negative staining TEM. While freshly isolated samples mostly contained individual pili, with a small fraction of pilus pairs, samples incubated for over a week contained a significant portion of pilus pairs as well as a few multi-pilus superstacks (Fig. 4a, b). After one month, the samples formed a gel-like structure (Supplementary Fig. 5a), comprising a dense network of flat Csu pilus stacks (Fig. 4c, d). The structure of these stacks and the architecture of the network formed in vitro closely resembled those observed between bacteria in 3D biofilms (Figs. 1–3, Supplementary Figs. 1, 2) with the Csu pili arranged into flat stacks of variable width that gradually merge into central structures comprising tens of pili.

**Fig. 2 | Csu pili form a network of stacks interlinking bacteria in 3D biofilms. a** *A. baumannii* 19096 T cells from liquid cultures are mostly covered in separate Csu pili pointing radially in every direction. **b–i** A negative stain TEM image of a suspended *A. baumannii* colony reveals a network of Csu pilus stacks (**b**), including local stacks wrapping around the bacteria (**c–e**) and a long, fully exposed superstack (**f–i**) linking two bacteria on the left to a clump of bacteria on the right, which they likely detached from during sample preparation. Typically, the pili growing in the narrow gaps between bacteria wrap closely around each bacterium (**c**) by forming pairs and flat stacks (**d**, **e**) that further merge into extensive networks connecting multiple bacteria (**b**, **f**). The ends of the exposed superstack connecting to the two groups of bacteria consist of Csu pilus pairs and stacks emerging from multiple directions (**g**, **i**) that then merge into a unified stack closer to the centre (**h**). Blue dots highlight examples of separate pili, while red dots border pilus pairs or stacking networks, and blue lines in (**d**) indicate individual pili in a multi-pilus stack. Diamonds in (**e–i**) highlight the local networks of pilus stacks merging into the superstack. Representative images were selected based on the quality of stain from micrographs recorded on 21 sample grids.

The striking similarity between the in vitro and in vivo networks, along with the progressing nature of their assembly observed in vitro, suggests that 3D biofilm formation is a spontaneous process inherently programmed within the structure of the pili.

## Structure of Csu pilus stacks

Csu pili from *A. baumannii* may have post-translational modifications in regions that can potentially affect both Csu pilus architecture and stack formation. To explore this possibility, we determined the rod structure of isolated *A. baumannii* pili at 3.28 Å resolution using single particle (SPA) cryo-EM (Fig. 5a, Supplementary Fig. 6a–d, h and Supplementary Table 1). The native pilus rod has the same symmetry and clinch structure as the recombinantly produced pili (Supplementary Fig. 6d), supporting our previous findings[22]. The resolution and quality of the map were sufficient to rule out any repeated post-translational modifications, suggesting that stack formation depends on the inherent structure of the Csu pilus itself.

Pilus pairs, representing the minimal pilus stack, are common within the 3D biofilms. Their relative abundance in the examined in vitro pili samples allowed us to select enough images for SPA cryo-EM structure determination. Since the pilus pair lacks helical symmetry, we generated an initial model based on the structure of the single pilus rod by optimising the overlap between its projections and a 2D class average (Supplementary Figs. 14–16). A filament-based SPA reconstruction yielded an 8.91 Å resolution density map, into which we fitted the model using rigid body real-space refinement (Fig. 5b, Supplementary Fig. 6e, i and Supplementary Table 1). The two Csu pilus rods form an antiparallel pair connected laterally by binding junctions at their zigzag corners (Fig. 6, Supplementary Video 3).

To determine the structure of multi-pilus stacks, we used a sample of isolated pili incubated for extended periods (Supplementary Fig. 5). We produced the cryo-EM maps of two alternative stacking architectures with the 7.58 Å map of type 1 stack showing three 18-subunit rod fragments, and the 4.70 Å map of type 2 stack showing four 17-subunit rod fragments (Fig. 5c, d, Supplementary Fig. 6f, g, j, k). Similar to the Csu pilus pair, the adjacent pilus rods in both types of stacks run antiparallel to each other, ensuring that contacts form primarily between pili from interacting cells rather than among the pili extending from the same cell (Fig. 6, Supplementary Videos 4, 5).

Although pilus pairs exhibit more flexibility with smooth bending, the structure of the rods and their contacts appear to be identical between the pair and the two types of stacks (Fig. 5b–d). Furthermore, these structures show only marginal differences from the structure of a single pilus rod (Fig. 5a), and to model the stacks, the helical twist of the native Csu pilus rod structure had to be adjusted less than 0.3°, representing less than a 0.2% change. This suggests that interacting pili can form contacts without significant induced fit in their overall structure, which may explain their high propensity to form stacks.

As with the single pilus, the Csu pilus rods in the stacks have a 188.66 Å long repeat with 3-turns-per-7-subunits symmetry, giving the pilus a heptagonal end projection with seven potential binding sites at 51.43° angles between them (Fig. 6, green and magenta arrows around the single pilus and pair, see also Supplementary Fig. 7a, b). The binding junctions in an interacting pilus pair form on the plane of their helical axes, which are 52.35 Å apart. Their bulk prevents a third pilus from binding at a 51.43° angle from the first junction (magenta arrows), but it can still bind at 154.29° and 102.86° angles (green arrows), corresponding to type 1 and type 2 stacks, respectively. The 154.29° angles make the type 1 stack slightly corrugated with 66.33 Å thickness and junctions at 12.86° angles to the underlying surface, whereas 102.86° angles generate the more corrugated type 2 stack, which is 84.58 Å thick with 38.57° angles to the surface (Fig. 6).

Both types of stacks consist of identical pilus rod pairs. However, adjacent pilus pairs are shifted relative to each other by one subunit in type 1 stacks and three subunits in type 2 stacks, respectively (Fig. 6, side views). Consequently, in type 1 stacks, the binding junctions cluster into lanes perpendicular to the rod axes and separated by four uninvolved subunits. In contrast, junctions in type 2 stacks are more evenly distributed along the pilus axes, with each junction separated from the next by two uninvolved subunits on one side and one subunit on the other.

The binding junctions consist of two subunit pairs, one from each adjacent, antiparallel rod (Figs. 5–7). The 4.70 Å resolution density map for the type 2 stack (Fig. 7a) and the 3.28 Å resolution structure of a single native Csu pilus (Fig. 5a) enable accurate modelling of the main chain and most side chains involved in the contact (Fig. 7b). The junction exhibits two-fold symmetry due to the complementarity of the two identical sides. The binding junction consists of a central region with two Thr14 and two Gln142 residues forming hydrogen bonds, and of two distal regions with Thr66 forming hydrogen bonds with Asn87 and Gln142 as well as hydrophobic interactions with Thr88, Ala89, and Gly143. Additionally, Ser120 may also interact with Ala89, though, being part of the flexible D″–E1-loop, its orientation is less restricted. The distal regions constitute a larger portion than the central region of the total interactive surface of ~415.95 Å² per junction. The charged residues around the binding site are too distant to form strong electrostatic interactions (Fig. 7c).

We next examined the contribution of individual residues to stack formation. Most mutations designed to have an effect at the pilus stacking interfaces also severely disrupted pilus assembly (Supplementary Table 3). However, the double mutant T66D–A89D, designed to introduce repulsive negative charges, and its variant T66D–ΔP68–A89D, featuring a shortened B″–C1 loop to position Asp66 closer to Asp89 and to prevent Asp66 from forming hydrogen bonds with the main-chain of the opposing Csu pilus (Supplementary Fig. 8a, b), were well tolerated. Although these mutations did not abolish antiparallel interactions in vivo, the more sensitive in vitro assays using purified pili showed that the mutants had smaller, less organised stacks with shorter continuous stacked regions compared to the wild type (Supplementary Fig. 8c, d), consistent with our structural data.

Sequence analysis of the interactive regions in CsuA/B subunits from 29 *Acinetobacter* species revealed two distinct groups (Supplementary Fig. 9a, Supplementary Fig. 10). The first group, comprising 18 sequences, differs from the A. *baumannii* CsuA/B sequence only by point mutations. The second group, consisting of nine sequences, all have a characteristic single residue deletion in the B″–C1-loop and a two-residue deletion in the C2–D-loop, both central to the binding junction. Modelling the junction for a member of the first group, with several point mutations to its interactive residues, resulted in similar

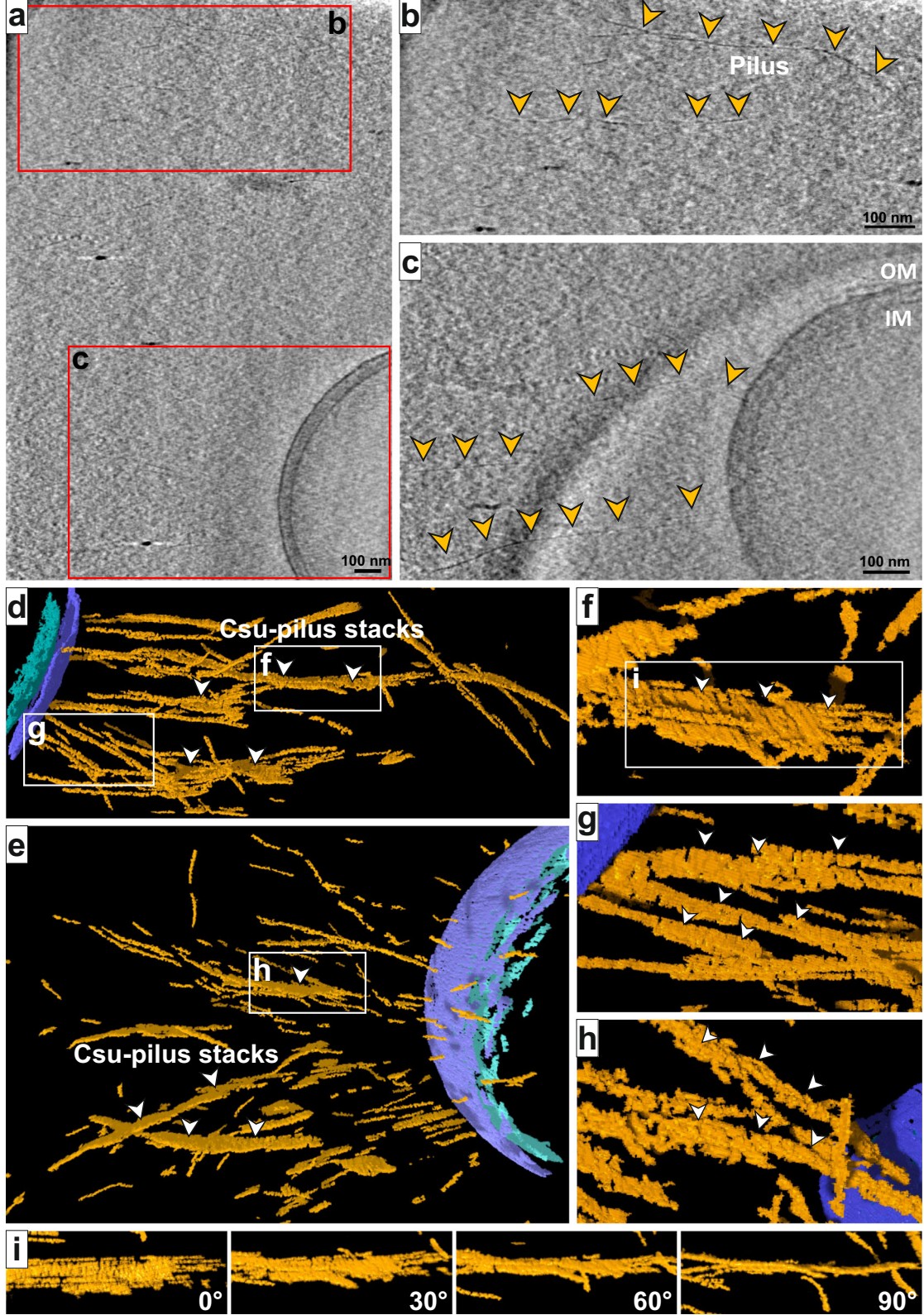

**Fig. 3 | Cryo-electron tomography shows flat Csu pilus stacks linking *A. baumannii* cells. a–c** A representative tomographic slice depicting cell-cell inter-actions in a fresh, suspended *A. baumannii* colony grown at 22 °C (**a**) and close-up views (**b, c**) (red rectangles) with 20–25 Z-slices averaged to show cross-sections of the Csu pilus stacks, as indicated (yellow arrowheads). **d, e** 3D segmentation of (**b**) and (**c**), highlighting the Csu pilus stacks (white arrowheads) mediating cell-cell contacts. The inner and outer membranes are in cyan and violet. **f–h** Zoomed-in views of the rectangular regions in the 3D segmentation images (**d, e**) showing flat Csu pilus stacks (white arrowheads) with individual pili being partially visible. **i** Rotating the view 90° around a stack in 30° increments shows the dif-ference between its thickness and width. Each stack is multiple times wider than they are thick, while stack thickness is always close to the width of an individual pilus, indicating that the stacks are typically flat. See also Supplementary Videos 1, 2.

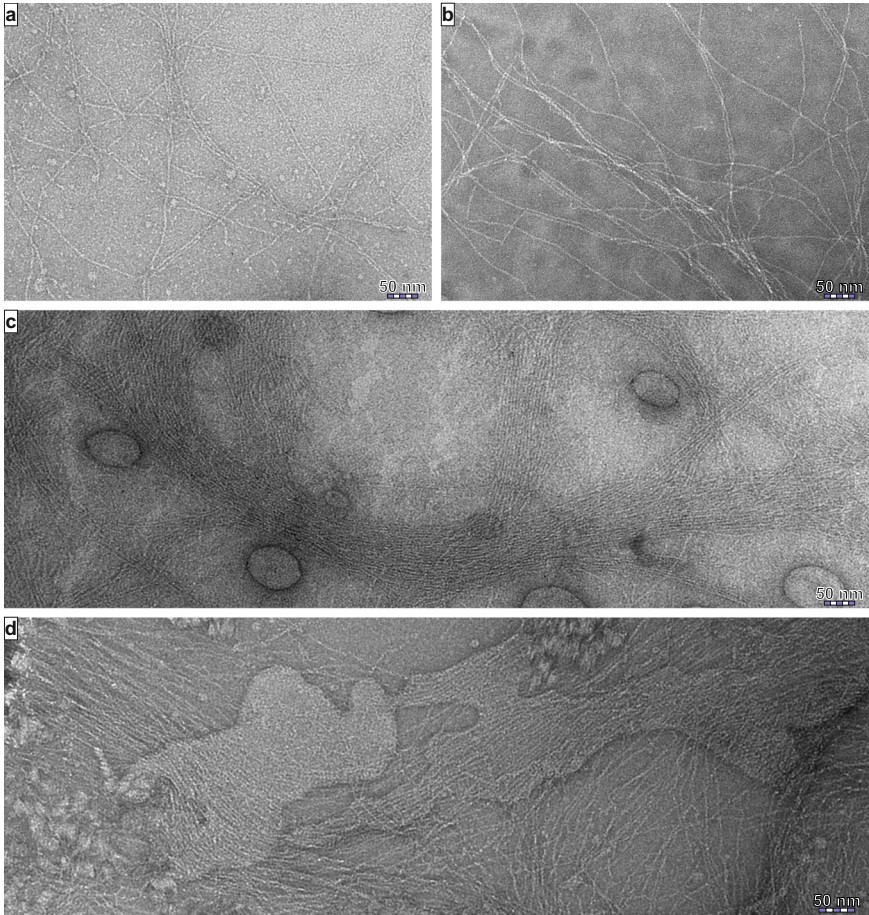

**Fig. 4 | Csu pili self-assemble into a network of stacks.** Negative stain transmission electron microscopy (TEM) micrographs of Csu pili extracted from *A. baumannii* 19096 T (**a**, **c**) or from *E. coli* carrying the Csu pilus expression plasmid pBAD-Csu (**b**, **d**). Liquid phase extracts have a high tendency to form long pilus pairs, which often merge into extensive superstacks (**a**, **b**). Extracts stored at +4 °C form complex networks of interconnected stacks that settle over time and form a gel at the bottom of the sample tube (**c**, **d**). Pilus pairs were observed on 14 sample grids with liquid phase samples, while superstacks were imaged on four grids prepared from gels in separate sample tubes. Representative images are shown.

interactions (Supplementary Fig. 9b, c). In contrast, the conserved deletions and point mutations in the second group created a uniform 2 Å gap throughout the antiparallel junction in the models (Supplementary Fig. 9d). However, manually moving the rods closer led to a tight contact due to surface complementarity. Thus, despite relatively low sequence conservation of the interactive residues, the contacts between rods appear to be conserved at the structural level, consistent with the considerable resilience of the junction to mutagenesis.

For more distantly related archaic CUP pili, the modelling of antiparallel binding becomes more challenging due to the uncertainty in the angle between subunits forming the zigzag architecture. However, this was possible for CupE pili from *P. aeruginosa*, as the structure of a single CupE pilus rod has been determined (PDB: 8CIO)[34]. Like Csu pili[16], CupE pili are necessary for *P. aeruginosa* to form mushroom-like 3D biofilms[15]. Furthermore, they also appear to form mesh-like arrays in vitro[34]. Interestingly, in the published cryo-EM density map (EMDB: EMD–16683) of a single CupE pilus rod, the C2–D-loop is linked to an unexplained density that disappears when the loop is mutated[34]. Its position indicates it could be part of an antiparallel CupE pilus included in the helical reconstruction, which allowed us to model the potential antiparallel junction (Supplementary Fig. 11b). Only a small adjustment to the helical twist (−1.33°, <1%) was necessary to generate the perfect 2-turns-per-5-subunits repeat capable of assembling into regular stacks (Supplementary Fig. 11c, d).

CupE has even shorter repeats (165.57 Å) than Csu pili (188.66 Å), as it consists of only five subunits, two of which our model predicts to be engaged in a binding junction (Supplementary Fig. 11b). Such unusually short repeats are possible only because of the zigzag architecture of the pilus, which generates a high density of binding junctions. For example, in a typical stack observed in vivo, a traced Csu pilus ran ~792 nm antiparallel to adjacent pili forming ~42 binding junctions with a predicted total interactive surface of around 17470 Å² (Supplementary Fig. 1b). Therefore, it is plausible that the archaic CU pilus zigzag architecture, which enables the formation of flat stacks, evolved to shape bacterial communities.

## Poly-N-acetylglucosamine and eDNA embed pilus stacks in mature biofilms

To study the kinetics of pilus stack assembly in vivo, we performed SEM analysis of *A. baumannii* biofilm formation on glass slides at different time points. Stack formation was unexpectedly fast, progressing simultaneously with 3D biofilm growth, and the stack network was essentially complete within 24–72 h (Fig. 1). However, over the course of days, we observed the stack network gradually becoming embedded in a less structurally defined material (Fig. 8a–d). This observation is consistent with previous findings that *A. baumannii* biofilms tend to accumulate significant amounts of PNAG and eDNA[14,16]. Higher resolution analysis using negative stain TEM revealed a close association of the pilus stacks with this extracellular substance, which essentially enwraps the individual stacks (Fig. 8e). Gold-labelled wheat germ agglutinin (WGA), extensively used to detect bacterial PNAG capsules, specifically binds to this

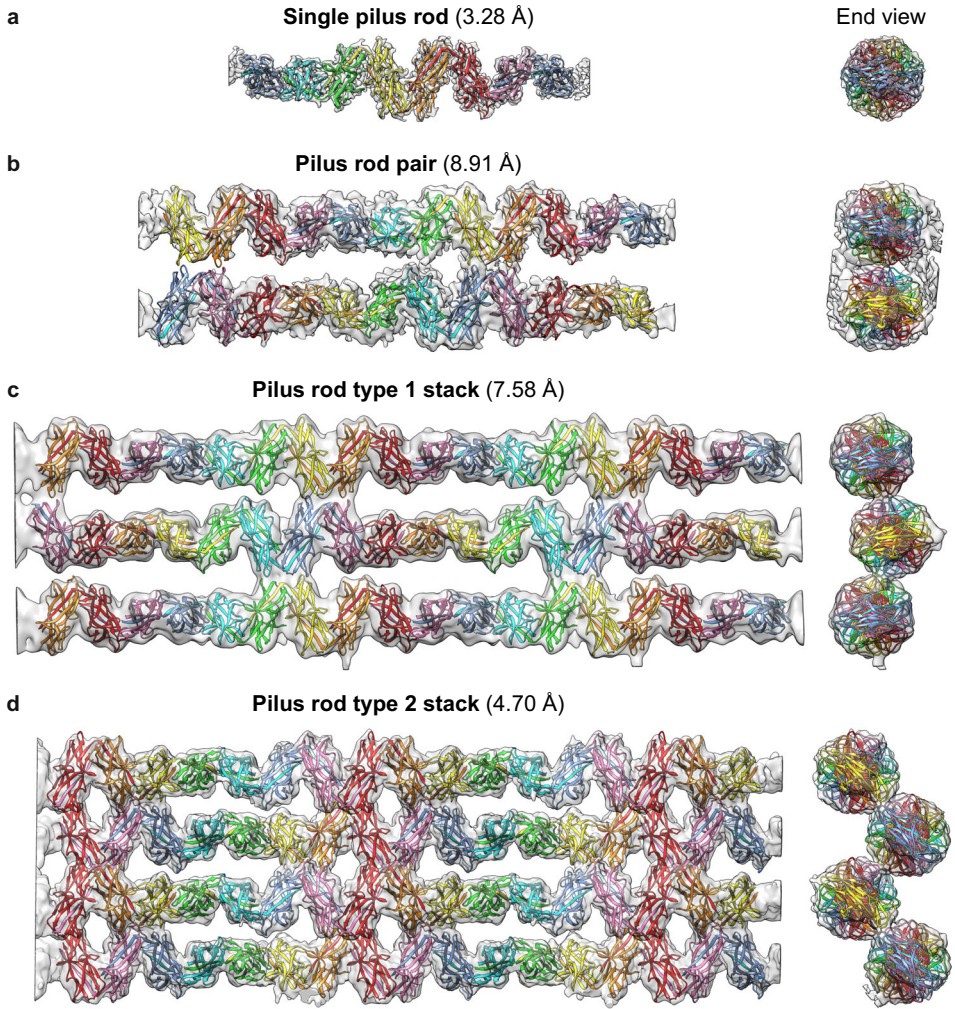

**a**  **Single pilus rod** (3.28 Å)                                              End view

**b**  **Pilus rod pair** (8.91 Å)

**c**  **Pilus rod type 1 stack** (7.58 Å)

**d**  **Pilus rod type 2 stack** (4.70 Å)

**Fig. 5 | Cryo-electron microscopy structures. a** Single *A. baumannii* Csu pilus rod. **b** Antiparallel pair of interacting pilus rods. **c** Csu pilus rod type 1 stack. **d** Csu pilus rod type 2 stack. The seven subunits in the helical repeat of the Csu pilus rod all have different colours. The map contour levels for the single pilus, pilus pair, type 1 stack and type 2 stack are 0.075, 0.37, 0.06 and 0.06, respectively. The resolution for each structure is shown in parenthesis.

substance, suggesting that it indeed contains a significant amount of PNAG (Fig. 8f, g).

While most bacteria seen in our negative stain TEM samples formed dense clumps with aplenty of Csu pilus stacks on the surfaces and PNAG or eDNA filling the tight gaps (Fig. 9a–c, Supplementary Fig. 2), others formed loose, yet cohesive pilus-linked rafts with otherwise empty, wide gaps (Fig. 9d–f). The distinctive lack of PNAG and eDNA in these rafts indicates that these EPS contribute little to clump cohesion. To test this further, we grew *A. baumannii* AB5075 mutant *pgaC::T26*, which is unable to produce PNAG[37], and treated suspended colonies with DNase I and the DNA-intercalating agent chloroquine to degrade eDNA[38]. Subsequent dilution and washing steps revealed several highly organised rafts, with tight, clear gaps and extensive Csu pilus stacks traversing them (Fig. 9g–i), further supporting the conclusion that Csu pilus stacks link the bacteria in early 3D biofilms independently of PNAG and eDNA.

**Mechanism of Csu pilus-mediated 3D biofilm formation**

Formation of pilus stacks in 3D biofilms is much faster than the assembly process observed in samples of purified pili, which may take months to complete. This suggests that something facilitates this process in vivo. We observed that in liquid cultures, the Csu pilus stacks are common at cell division zones as well as at the ends of bacterial rods, possibly as remnants of cell divisions (Supplementary Fig. 3c–e). During cell division, the cell membrane grooves inward around the septum, positioning Csu pili around the septum antiparallel to each other, which seems to promote pilus stack formation (Fig. 10). Connecting daughter cells by pilus stacks is likely a major step in the formation of a 3D biofilm, allowing the stacks to form simultaneously with bacterial growth (Supplementary Video 6).

Although this process can occur both in liquid and solid media, the formation of stacks linking bacteria together is dramatically more efficient in 3D biofilms than in liquid cultures. *A. baumannii* in biofilms produce more pili than liquid culture cells[39], but since clumps are also rare in *E. coli* liquid cultures overexpressing Csu pili, the higher number of pili is unlikely to be the main factor here.

Potentially, bacterial attachment to surfaces can promote subsequent formation of 3D biofilms, even though these two processes can occur independently, mediated by different parts of the pili (Fig. 10, Supplementary Fig. 4e, f). First, the stacks formed between daughter cells attached to a surface would experience lower levels of turbulence, therefore preserving the contacts. In contrast, *A. baumannii* cells growing in a liquid medium are less likely to clump, probably because turbulence, moving both Csu pili and the cells around, inhibits pilus stack formation and breaks most pairs that form (Supplementary Figs. 3, 4 and 12). Second, the narrow gaps between neighbouring cells in a 3D biofilm help position Csu pili antiparallel to each other, guiding the formation of extensive Csu pilus stacks

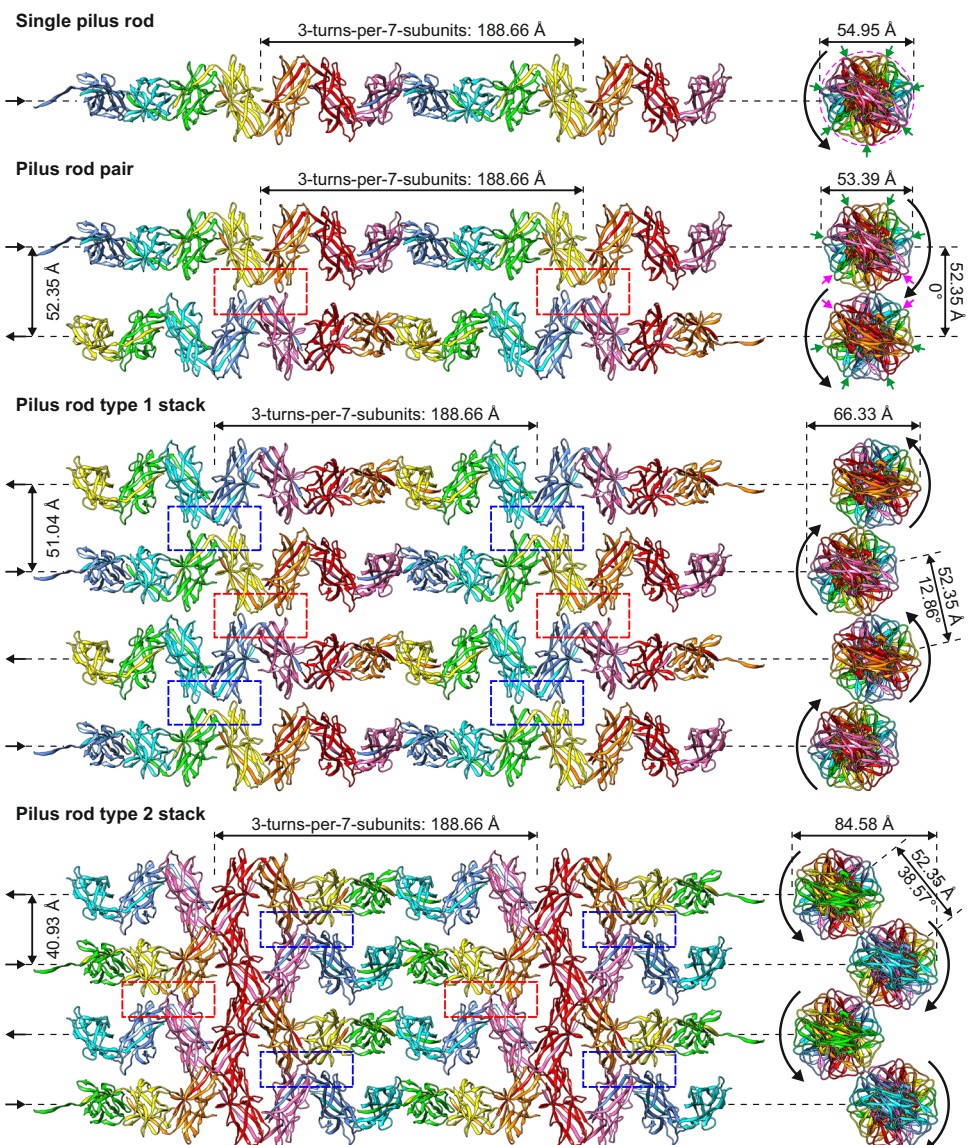

**Fig. 6 | Pilus rods assemble into flat antiparallel stacks.** Cartoon diagrams of 14-subunit fragments of a single Csu pilus rod, an antiparallel pair and type 1 and 2 stacks. The seven subunits in the helical repeat of the Csu pilus rod all have different colours. The straight and curved arrows point towards the direction of the outer membrane and indicate the rod helical twist, respectively. The red and blue rectangles mark the binding junctions viewed from opposite directions. The heptagonally distributed green arrows around the end views of a single pilus rod and a pair mark the available binding sites, whereas the magenta arrows mark sites blocked by a pilus occupying a neighbouring site. See also Supplementary Videos 3–5.

(Figs. 9, 10). Such extensive stacks between daughter cells and neighbouring cells could help maintain their spatial orientation, allowing the biofilm to grow taller without breaking apart. In liquid cultures, such neighbouring cells are most often absent (Supplementary Fig. 12), unless the liquid culture is stagnant, making its conditions similar to solid media and allowing the formation of a surface pellicle with CsuA/B as its most abundant protein component[36]. Finally, our results show that 3D biofilms on solid media accumulate PNAG, eDNA, or other secreted materials that fill the narrow gaps between the cells, further fixing cell positions and stabilising the pilus stack network (Figs. 8, 10).

## Discussion

We reveal the key mechanism explaining how the antibiotic-resistant 3D bacterial biofilms of *A. baumannii* form. This involves a network of pilus-pilus contacts that develops simultaneously with bacterial growth, interlinking and fixing bacterial cells in three-dimensional space. Formation of this network depends on a novel superstructural feature that we denote "flat pilus stacks", in which pili pack into antiparallel sheets with their rods connected laterally by junctions at their zigzag corners. We show that such flat assemblies require helical filaments with very short repeats, suggesting that the zigzag architecture of archaic CUP pili has likely evolved to mediate 3D biofilm formation. Although bacterial clumping in 3D structures and bacterial adhesion on surfaces use different parts of the Csu pili—the pilus rod and hydrophobic tip fingers, respectively—our results suggest that these, in principle, independent processes may greatly facilitate each other during 3D biofilm formation.

Individual Csu pili are characterised by superelasticity, a unique property that allows them to stretch reversibly to twice their original length at a constant tensile force (Supplementary Fig. 13a)[22]. Interestingly, our modelling suggests that this property is largely preserved in the flat stacks (Supplementary Fig. 13b–d). In a pilus rod pair, two in every seven-subunit helical repeat would need to maintain the interactions, leaving five subunits free to extend (Supplementary Fig. 13b, Supplementary Video 7). In Csu pilus type 1 stacks, the binding junctions overlap, forming interlocked three-subunit lanes separated by

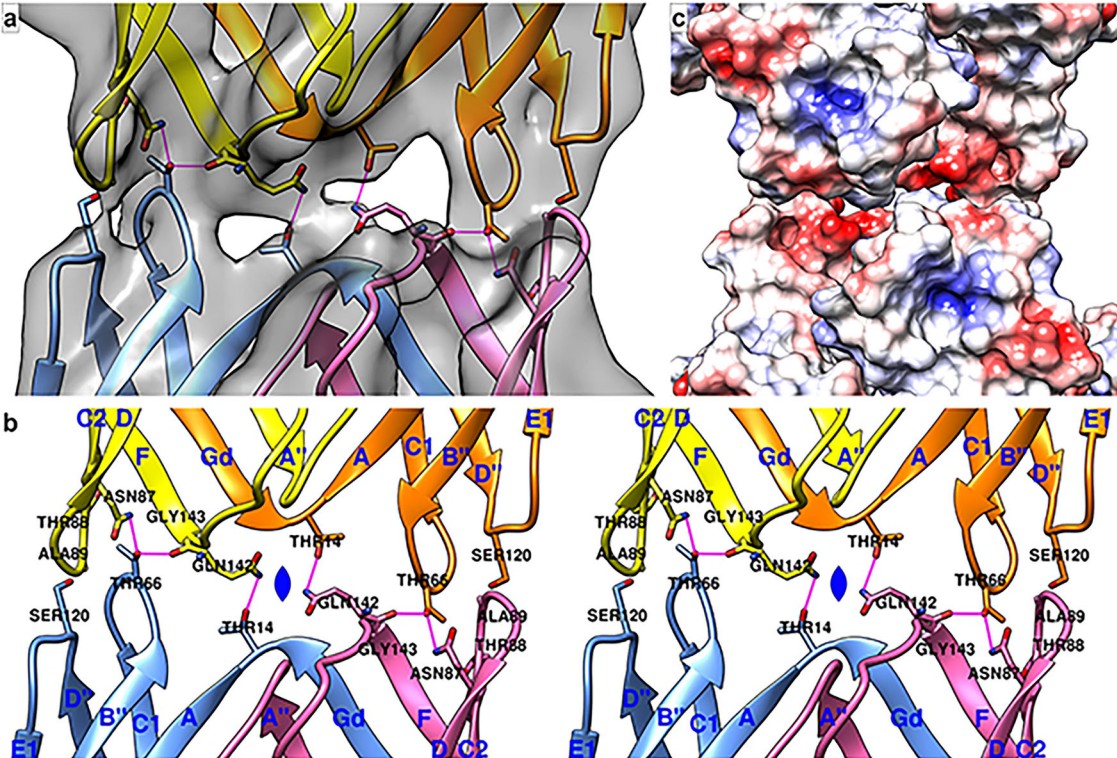

**Fig. 7 | Structure of the Csu pilus antiparallel rod binding junction. a** A fragment of the Csu pilus type 2 stack structure, featuring two adjacent subunits from one pilus rod (yellow and orange) forming a binding junction with two adjacent subunits from another rod (pink and blue), fits within a contour level 0.064 density map. The residues shown are those indicated by the density map to interact across the binding junction, with magenta lines representing predicted hydrogen bonds. **b** Cartoon diagram of the junction between pilus rods (stereo view). The subunits are equivalent to the ones in (**a**), with the two-fold symmetry of the junction indicated. Side chains forming hydrogen bonds or hydrophobic interactions are shown in sticks and labelled. **c** Electrostatic potential surface of the junction. Regions of positive surface potential are depicted in blue and negative potential in red (scale is from −1 to 1 kT).

four uninvolved subunits that could stretch along the rod axes (Supplementary Fig. 13c). In contrast, in type 2 stacks, each binding junction is flanked along the rod axes by one or two uninvolved subunits separating it from its neighbours. With only three uninvolved subunits per repeat, type 2 stacks are expected to be less extendable along the rod axes. However, without shared subunits interlocking the binding junctions, the type 2 stack may also stretch perpendicular to pilus axes (Supplementary Fig. 13d).

This extensive stretching capability may explain why Csu pili tend to form only flat stacks, even though the available binding sites theoretically allow for much greater variation (Supplementary Fig. 7a, b). It appears that, apart from the wavy type 1 stack variant that may have a somewhat reasonable ability to stretch (Supplementary Fig. 7c), the spacing of binding junctions in others would exclude most of the subunits from stretching (Supplementary Fig. 7d–f). Importantly, the high elasticity of stacks would make established 3D bacterial clumps more resistant to turbulent environments by absorbing stress. Interestingly, the stacks embedded into extracellular material made of PNAG and eDNA are reminiscent of reinforced concrete, as in this combination, the stack may act like steel bars or mesh, providing resistance to tensile and shear stresses, while PNAG or eDNA would provide resistance to compression. Further experimental biophysical studies will be required to investigate the predicted properties.

Csu pilus stack formation is a surprisingly rapid process, occurring simultaneously with 3D biofilm growth. Our results suggest that the fast kinetics may be explained by the initiation of pilus stacking around the septa of dividing cells. Pilus–pilus contacts may form quickly through a zipper-like mechanism, in which the establishment of each binding junction facilitates the formation of the next. The high

cooperativity of this process may depend on the relatively flexible tips of the B″–C1 and D″–E1 loops (Supplementary Fig. 6c), which, according to our bioinformatic analysis and mutagenesis data, can accommodate a range of binding geometries. Further studies will be needed to clarify this mechanism.

Our results support the dominant role of archaic Csu pili in clumping bacteria into 3D structures. However, a role for other EPS, such as PNAG and eDNA, at different stages of this process cannot be excluded. Similarly, several protein surface receptors, including OmpA[40], Ata[41], as well as Biofilm-Associated Protein (Bap) and its closely related Blp1 and Blp2[42,43], facilitate biofilm formation. As these receptors have smaller structures, they likely mediate more localised interactions compared to the Csu pili. In addition, *A. baumannii* produces Type IV pili, and its different variants promote motility in the AB5075 strain and biofilm formation in the ACICU strain[44]. Interestingly, the latter strain has a transposase gene inserted between *csuA/B* and *csuA* within its *csu* operon. In our colonies of *A. baumannii* 19096 T and AB5075, we observed little to no filaments wide enough (72 Å)[45] to be interpreted as Type IV pili. Only colonies grown at 30 °C and 37 °C displayed a few bacteria with such filaments, while most cells expressed abundant Csu pili. This agrees with the observation that cyclic AMP acts as a master regulator, suppressing the expression of Csu pilus subunits, PNAG-synthesising enzymes and quorum sensing, and thus biofilm formation in general, while strongly upregulating Type IV pilus subunits and other motility-related genes[46]. Therefore, while Type IV pili may contribute to biofilm formation by mediating adhesion to certain surfaces, they likely lose relevance once *A. baumannii* switches to producing Csu pili. However, the expression regulation could be different in other

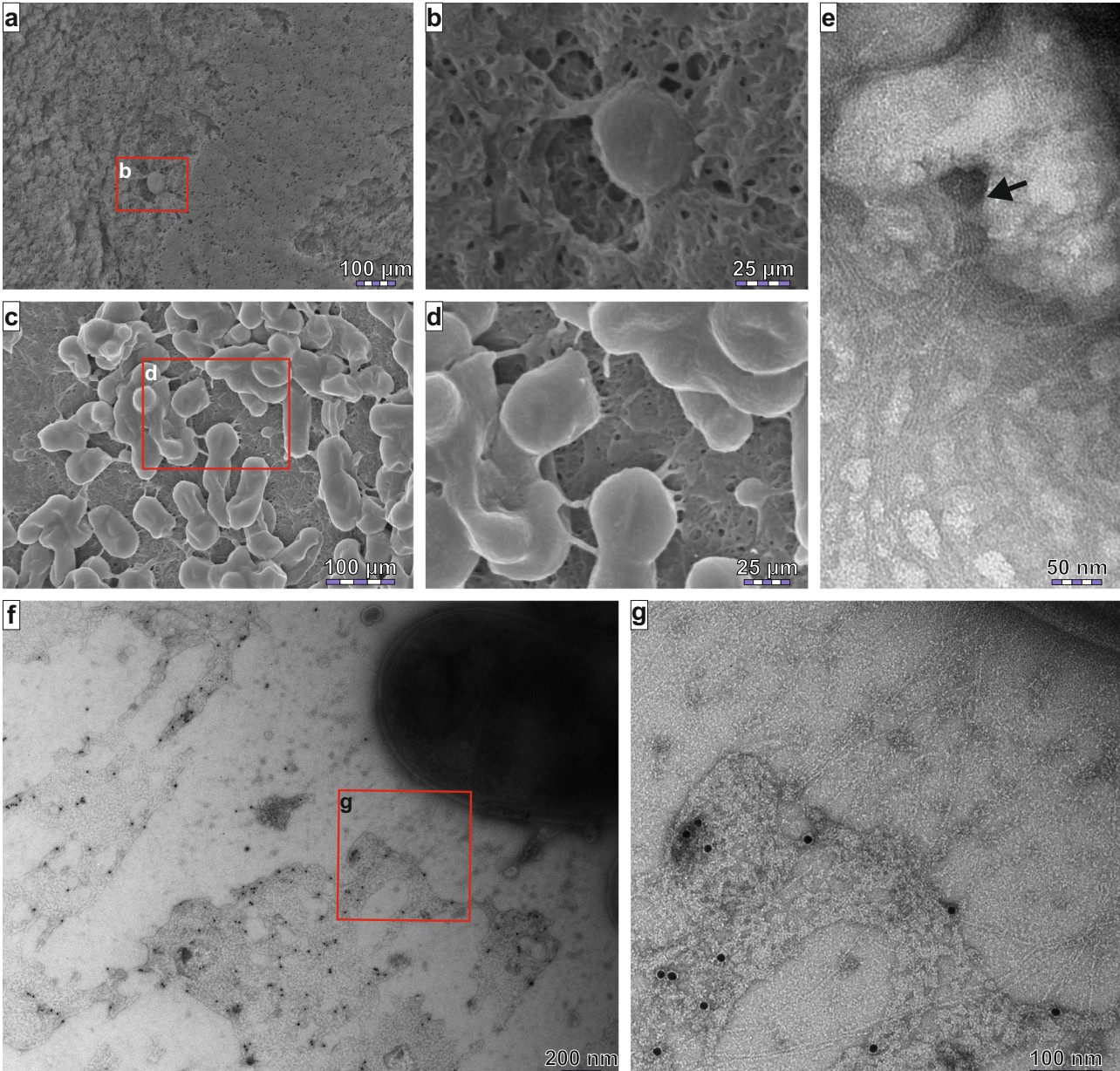

**Fig. 8 | Csu pilus-mediated 3D biofilm is sealed by PNAG. a–d** SEM images of *A. baumannii* AB5075 (**a**, **b**) and *A. baumannii* ATCC 19606 T (**c**, **d**) grown at 30 °C for 10 days before samples were fixed and prepared for SEM analysis as described in "Methods". Areas indicated by red rectangles in (**a**) and (**c**) are shown at a higher magnification in (**b**) and (**d**), respectively. Representative SEM images were selected from micrographs of grids prepared from three individual colonies of each strain. **e** Negative stain TEM image of bacteria from a suspended *A. baumannii* strain CCUG 19096 T colony showing interactions between Csu stacks and PNAG at a higher resolution. The arrow points at a position where the Csu pilus stack emerges from underneath PNAG. **f** *Triticum vulgare* lectin (WGA) colloidal gold conjugate binds almost exclusively to the amorphous mass surrounding the bacteria from a suspended *A. baumannii* colony, which indicates the mass contains PNAG. Suspending the mature *A. baumannii* colony in WGA colloidal gold conjugate solution has dispersed PNAG away from the bacteria. **g** Close-up image showing WGA colloidal gold conjugates bound to PNAG near partially stacked Csu pili extending from the bacterial surface. TEM images of WGA colloidal gold conjugate bound to PNAG were selected from micrographs of five grids prepared from individual *A. baumannii* strain CCUG 19096 T colonies.

species. Studies of other archaic CUP pili, such as the *P. aeruginosa* CupE system, may help clarify this.

Archaic zigzag pili represent the first example of using highly structured, regular flat sheets for 3D biofilm assembly. Several other EPS scaffolds reported in bacteria and archaea are also DSC-linked fibres, but they mostly form twisted[47,48] or cable-like bundles[49–52]. *E. coli* common pili, assembled through classical CUP mechanisms[47], and archaeal bundling pili (ABP)[51] from *Pyrobaculum calidifontis* involve antiparallel interactions, while archaeal Tafi pili from *Natrinema* sp. J7-2 appears to involve both parallel and antiparallel interactions[52]. The directionality remains unknown for *Acinetobacter*

sp. strain BD413 classical Acu pili[48], *Sulfolobus acidocaldarius* archaeal thread filaments[50] and TasA pili of the Gram-positive bacterium *Bacillus subtilis*[49]. Further studies are necessary to understand how these bundles spontaneously assemble to support 3D biofilms.

Considering the important role of bacterial 3D associations in protecting pathogens from antibiotic treatment, it would be reasonable to treat infections with inhibitors of 3D association along with antibiotics. Thus, our findings suggest that methods to inhibit pilus-pilus interactions might represent a new strategy to fight multidrug-resistant bacterial infections.

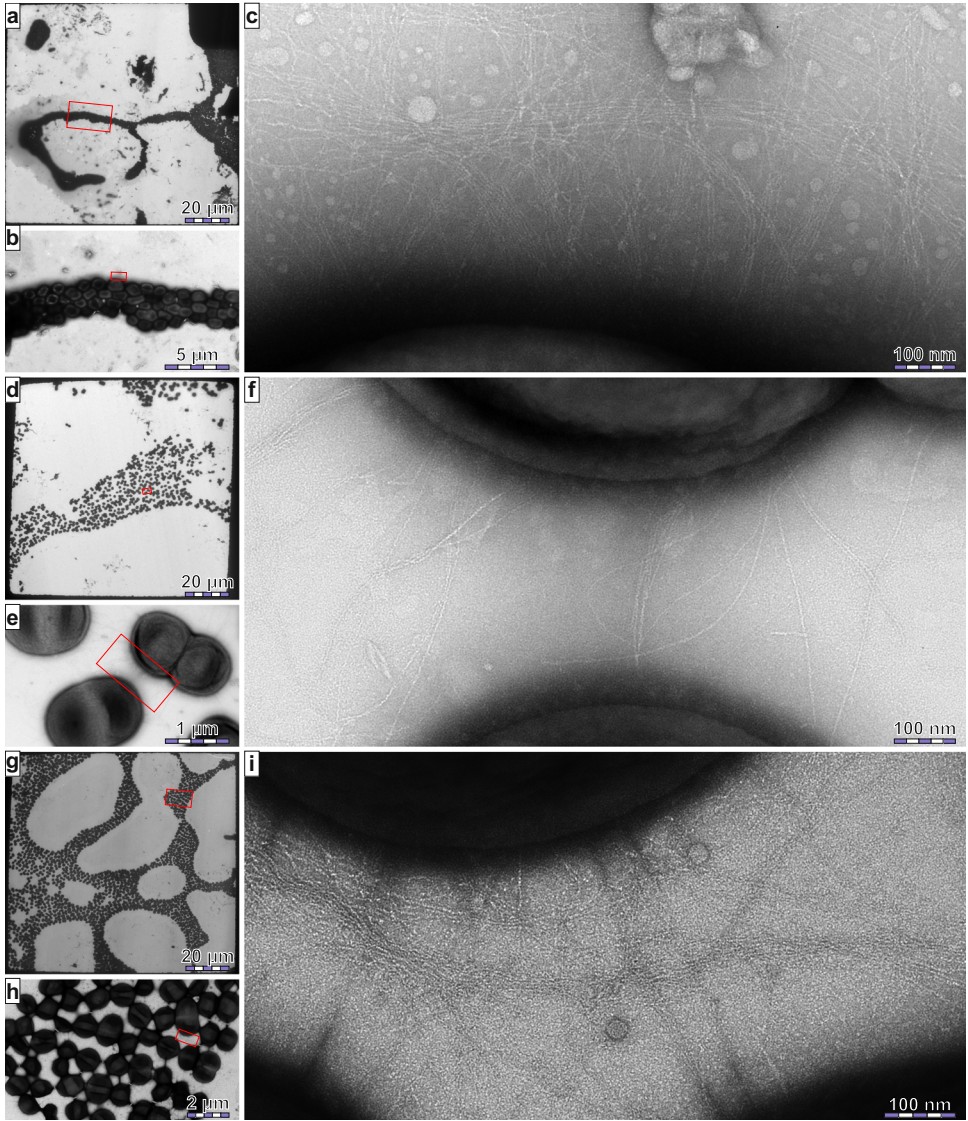

**Fig. 9 | Csu pilus stacks maintain cohesion in *A. baumannii* strain AB5075 colonies independently from PNAG and eDNA. a, b** Transmission electron microscopy images of suspended *A. baumannii* strain AB5075 wild type colonies show that the bacteria mostly settle on the grid as large cohesive clumps (**a**) with narrow gaps obscured by amorphous, well-staining substances, such as PNAG and eDNA (**b**), and aplenty of Csu pilus stacks on the outer surfaces (**c**). **d–f** Loose, but cohesive rafts of bacteria (**a**) appear occasionally on grids with suspended *A. baumannii* strain AB5075 wild type colonies, and while the wide gaps contain no amorphous materials (**e**), individual Csu pili, pairs and small stacks regularly cross the gaps (**f**), though in lower quantities than on the surfaces of bacterial clumps. **g–i** *A. baumannii* strain AB5075 mutant *pgaC::T26* lacks the ability to produce PNAG, and treating its suspended colonies with DNase I and DNA-intercalating agent chloroquine leaves highly organised tight rafts closely resembling the normal bacterial clumps (**a**), but with narrow, clear gaps (**g**), and extensive stacks similar to the ones on normal clumps (**c**) traversing the gaps (**i**). Colonies grew for three days at 22 °C, immediately followed by grid preparation. Clumps of wild-type *A. baumannii* cells with Csu pilus stacks and loose rafts were imaged on 19 grids, with 28 and 4 grid squares inspected for stacks, respectively. *A. baumannii* strain AB5075 mutant *pgaC::T26* with DNase I treatment was studied on 10 grids. Representative images are shown.

## Methods

### Production and purification of Csu pili from liquid cultures

Csu pili were extracted from either *E. coli* BL21-AI (F⁻ompT hsdSB(r<sub>B</sub>⁻ m<sub>B</sub>⁻) gal dcm araB::T7RNAP-tetA; Invitrogen) transformed with the pBAD-Csu plasmid[21] or the *Acinetobacter baumannii* CCUG 19096 T strain[53], which we selected for the translucent phenotype[54], as it is generally associated with a higher number of pili[35]. The BL21-AI preculture was grown overnight in LB medium containing 100 µg/ml ampicillin at 37 °C with shaking at 225 rpm. Subsequently, 7 ml of this preculture was inoculated into 300 ml of LB medium with 100 µg/ml ampicillin and grown with shaking at 225 rpm, until an optical density (OD) at 595 nm of around 1.0 was reached. The shaking speed was then reduced to 150 rpm, and Csu pilus production was induced by adding 0.2% (w/v) L-arabinose for an additional 2.5 h. For the *A. baumannii* cells, cultures were grown without antibiotics in 300 ml of LB medium at 22 °C with shaking at 150 rpm for 26–28 h. The same procedures were applied when growing liquid cultures for negative stain transmission electron microscopy (TEM), except that liquid cultures used to observe bacterial clump formation were left stagnant for around 1 h at 22 °C before taking samples for TEM.

The extraction of Csu pili followed the same protocol for both sources. Cells were first centrifuged at 15,971 × *g* for 15 min. The resulting pellet was resuspended in 3.6 ml of 20 mM HEPES, 75 mM NaCl (pH 7.4), and the Csu pili were extracted by heating the suspension to 65 °C for 60 min. Following this, the mixture was centrifuged at 17,700 × *g* for 15 min, and the supernatant was collected and dialysed

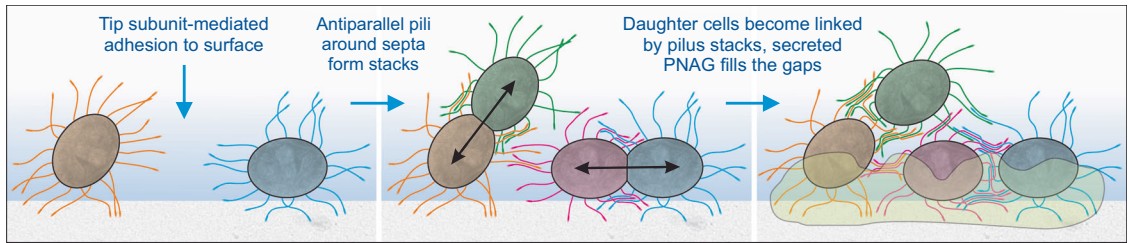

**Fig. 10 | Proposed mechanism of Csu pilus-mediated 3D biofilm formation.**
Biofilm formation starts with bacterial attachment to a surface, here mediated by the hydrophobic tip-fingers of the Csu pilus tip-subunit CsuE. Low turbulence enables bacteria to remain in close proximity long enough for a network of linking Csu pilus stacks to form around the septum of separating daughter cells and with the closest neighbouring cells on the surface (See also Supplementary Fig. 12, Supplementary Video 6). The Csu pilus stacks fix the relative positions of the bacteria, enabling them to remain in place even if they lack direct contact with the surface. Finally, PNAG, eDNA and other secreted materials fill the gaps between bacteria, providing compression resistance to the biofilm.

against 20 mM HEPES, pH 7.4. The sample was then filtered using either a 0.22 µm or 0.45 µm filter and applied to a column packed with 20 ml of Source 30Q (Cytiva), using either the Äkta Purifier UPC 10 or Äkta Pure (Cytiva). Csu pili were eluted at a flow rate of 0.9 ml/min using a 50% gradient of 1.0 M NaCl in 20 mM HEPES, pH 7.4, over 50 min. Fractions with the highest concentrations of CsuA/B and least impurities were selected based on SDS-PAGE analysis and used without further purification.

### Growing *A. baumannii* and recombinant *E. coli* colonies for Csu pilus stack analysis

We initially cultivated *A. baumannii* CCUG 19096 T strain colonies using a ten-fold dilution series to achieve approximately 100–200 colonies per 0.5x LB agar plate (composed of 2.5 g NaCl, 5 g tryptone, 2.5 g yeast extract, and 8 g agar, pH 7.4, in 1 L of distilled water) to ensure the manifestation of the translucent phenotype. We later observed that even seemingly opaque colonies had enough phenotype-switched cells to produce large pilus-linked clumps of bacteria. For the early negative stain transmission electron microscopy (TEM) experiments, the bacteria were grown for 16–20 h at 37 °C until the colonies reached a diameter of approximately 1.0–1.5 mm, and then stored at 8 °C to keep them from growing further. Under this condition, colonies could be stored for 2–3 weeks, and remained usable even after three months, with the only apparent change being the slow accumulation of amorphous materials, presumably PNAG or eDNA. For the later TEM experiments using *A. baumannii* AB5075 strain colonies and for cryo-ET using *A. baumannii* CCUG 19096 T strain colonies, we grew them at 22 °C for three days before using them immediately for the experiments without any storage. Finally, for the TEM experiments focusing on stack formation in variable growth conditions, one variable was growing them at 30 °C or 37 °C for less than 18 h before preparing the grids. The other growth condition experiments grew for three days at 22 °C, but had pH adjusted to 8.5 or 6.0 instead of 7.4, or NaCl concentration of 200 mM or 400 mM instead of 42.8 mM. The assembly of Csu pili in recombinant *E. coli* colonies was induced by growing them for 15–17 h at 37 °C in a low concentration of L-arabinose (0.001% w/v) to limit the number of Csu pili produced.

### Single-particle cryo-EM

To determine the structure of the native Csu pilus rod, 3.5 µl of sample purified from *A. baumannii* CCUG 19096 T strain was applied to glow-discharged 300-mesh gold holey-carbon grids (Quantifoil R1.2/1.3, Electron Microscopy Sciences) and vitrified using a Vitrobot Mark IV (Thermo Fisher Scientific) at 4 °C and 100% humidity. The data were collected on a Titan Krios transmission electron microscope (Thermo Fisher) operating at 300 kV equipped with a K3 direct electron detector (Gatan) in counting mode with a BioQuantum GIF energy filter (slit width of 20 eV) at a magnification of × 81,000 corresponding

to a pixel size of 1.08 Å at the specimen level. 60-frame videos with a dose rate of -15 electrons per pixel per second and a total accumulated dose of 54.8 e-/Å² were collected using the Latitude-S (Gatan) single-particle data acquisition programme. The nominal defocus values were set from −0.8 to −2.5 µm.

To determine the structure of the Csu pilus pair and the initial structure of a Csu pilus stack, 4 µl of sample purified from *E. coli* BL21-AI with pBAD-Csu plasmid was applied to glow-discharged 300-mesh copper holey-carbon grids (Quantifoil R1.2/1.3, Electron Microscopy Sciences) and vitrified in liquid ethane using a Vitrobot Mark IV (Thermo Fisher) at 4 °C and 100% humidity. The data were collected on a 300 kV Titan Krios electron microscope (Thermo Scientific) equipped with a Gatan K2 Summit direct electron detector operated in super-resolution mode with a pixel size of 0.851 Å and a defocus range of −0.8 to −2.5 µm. A total dose of 50 e-/Å² was applied and equally divided among 40 frames to allow for dose weighting.

To determine the final Csu pilus stack structures, we used highly concentrated fractions of Csu pili, which we purified from *E. coli* BL21-AI with the pBAD-Csu plasmid and then stored at 4 °C for over a year. The sample comprised an upper liquid phase and a lower gel-like phase containing Csu pilus stacks (Supplementary Fig. 5a). 3 µl of the gel phase was applied to glow-discharged 300-mesh copper grids with a carbon film (Quantifoil R1.2/1.3, Electron Microscopy Sciences) and vitrified using a Vitrobot Mark IV (Thermo Fisher Scientific) at 4 °C and 100% humidity. The data were collected using a 300 kV Titan Krios electron microscope (Thermo Fisher Scientific) equipped with a Gatan K3 direct electron detector in super-resolution mode, using a pixel size of 0.825 Å and a defocus range of −0.4 to −1.6 µm. After manually picking the grid holes (Supplementary Fig. 5b–d), we used automated cryo-EM data collection utilising EPU v3.6.0 collection software (Thermo Fisher Scientific). The total dose was 59.848 e-/Å², split evenly across 45 frames, giving each frame a dose of 1.330 e-/Å².

### Cryo-EM image processing and reconstruction

The movie frames of the native Csu pilus rods were subjected to beam-induced motion correction using Patch Motion Correction in *CryoSPARC v4.5.3*[55] followed by determination of Contrast Transfer Function (CTF) parameters in Patch CTF. Micrographs with CTF fit better than 3.5 Å were used for further analysis. For the native Csu pilus rod, the particles picked with Filament Tracer (189,846 particles) were subjected to 2D classification followed by Ab Initio model generation with two classes. The best class from Ab Initio (50,502 particles) was used for non-uniform refinement, local motion correction and reference-based motion correction, generating a map with a global resolution of 3.28 Å (Supplementary Fig. 17a).

The movies of the Csu pilus pair were subjected to beam-induced motion correction using MotionCor2[56] followed by determination of CTF in CTFFind-4.1[57] in *RELION 3.0*[58]. Micrographs with CTF fit better than 3.5 Å were used for filament picking. The filament pairs were

manually picked in *RELION* (35,938 particles), extracted with a box size of 300 pixels and imported to *CryoSPARC*. The particles were then subjected to 2D classification, followed by Ab Initio model generation and homogeneous refinement. The final particle stack (12,883 particles) generated a map with a global resolution of 8.91 Å (Supplementary Fig. 17a).

The movie frames from the initial test set of Csu pilus stacks were processed using Patch Motion Correction and Patch CTF in *CryoSPARC v4.5.3*[55] for beam-induced motion correction and CTF estimation. Micrographs with a CTF fit better than 3.5 Å were selected for particle picking using Filament Tracer, yielding 136,628 particles. Only about a quarter of the filaments represented pilus stacks, with the remainder corresponding to single pilus rods. The particles were then subjected to 2D classification followed by Ab Initio model generation and homogeneous refinement, generating an 8.5 Å map (Supplementary Fig. 17a) that served as the initial model for processing a larger Csu pilus stack dataset (Supplementary Fig. 17b–h).

The movies from the larger dataset of Csu pilus stacks were subjected to beam-induced motion correction using Patch Motion Correction in *CryoSPARC v4.5.3*[55], followed by determination of Contrast Transfer Function (CTF) parameters in Patch CTF. In Manually Curate Exposures, we rejected 736 and left 7205 exposures. Filament tracer gave 32,866 particles after removing bad picks with Inspect Picks, Extract Micrographs and two rounds of 2D Classification and Select 2D Classes. The best 2D classes were used in template-based picking, which resulted in 367,173 particles after removing bad picks.

Homogeneous Refinement using our initial 8.5 Å map of the type 1 stack revealed that there are two types of stacks. We produced models of the two stacks to create 16 Å simulated maps in *UCSF Chimera*[59], and used *ChimeraX*[60] and Volume Tools in *CryoSPARC* to produce masks for two Homogeneous Refinements. Using the resulting maps to sort the particles in Heterogeneous Refinement revealed a need for more particles. A second round of template-based picking gave two mixed particle sets, which we sorted with three pairs of Heterogeneous Refinement using the previous two Homogeneous Refinement maps, and every time, combining the particles sorted for one type of stack in one group. This resulted in 710,036 particles for type 1 and 924,343 for type 2 stacks after the Remove Duplicate Particles jobs.

Both types of stacks had issues with pseudosymmetry due to pilus helical symmetry, and stacks always resembling a row of pillars, even if flipped 180° in any direction. This caused the binding junctions to disappear as the increasing number of flipped particles biased the search maps. Remnant particles representing the wrong type of stack further exacerbated the issue. We reduced the issue for the type 1 stack by splitting the particles into seven sets, refining each with Homogeneous Refinement, and excluding two sets with the weakest binding junction densities. The main solution to the pseudosymmetry issue was rotating a 16 Å simulated map 180° in every direction to generate seven search models to sort the particles in Heterogeneous Refinement based on their orientations. This required having few contaminant particles and deliberately lowering the O-EM learning rate, so that occasional flipped particles would not bias the maps. Re-orienting the search models with their sorted particles back to the initial orientation and running Local Refinement with the combined dataset, while limiting particle rotation, yielded the final maps. The Local Refinement for type 1 stack used 504,940 particles. We used a type 1 stack simulated map also as a decoy class, when sorting the particle orientations for type 2 stack, reducing its final number of particles used for Local Refinement to 687,088 (Supplementary Fig. 17b–h, Supplementary Table 1).

### Model building and refinement

To construct the model for the native *A. baumannii* Csu pilus rod, we fit subunits from our previous recombinant Csu pilus rod structure (PDB: 7ZL4) into the density map using the Fit in Map tool in *UCSF Chimera*. First, as previously[22], we adjusted the angle between two subunit models to match the average angle between two subsequent subunits in multiple positions along the pilus rod density map. Due to the map voxel size, this still left a small gap between the subunits, so in the second step, we shifted the two subunits closer in *Coot 0.8.9.2*[61] to link the lower subunit to its donor strand, and used the shifted pair to adjust the map voxel size in *Chimera*. We iterated these two steps until subunit positions stabilised. After further manual adjustments in *Coot*, the structure was refined using the rigid body and per-residue B factor real space refinement with *PHENIX* v1.20.1-4487[62].

The native pilus structure served as the basis for building models of the Csu pilus pair and both types of stacks, with minor adjustments to the helical twist. The angle between rod pairs in all structures was determined based on our overlap analysis with a 2D class (Supplementary Figs. 14–16, Supplementary Note 1). The binding junction was modelled using the 4.70 Å map of the type 2 stack, and this structure was replicated for the Csu pilus pair and the type 1 stack, as the experimental maps for these structures had lower resolution. The structures were refined as described above for the native Csu pilus structure. Refinement statistics are shown in Supplementary Table 1.

We determined the interactive surface of a binding junction using the CCP4i implementation of AREAIMOL[63] by first calculating the sum of individual surfaces from two antiparallel pilus fragments and then reducing from that the surface of the fragments combined in a single model.

### Cryo-ET data collection and processing

For cryo-ET sample preparation, *A. baumannii* colonies were suspended in a solution containing 10 nm colloidal gold beads (Sigma-Aldrich, Australia) pre-coated with 1% BSA. The mixture was applied to glow-discharged R2/2 Quantifoil holey carbon grids (Quantifoil Micro Tools GmbH, Jena, Germany). Freezing was performed using Vitrobot (FEI Thermo Fisher Scientific), maintained at 100% humidity. Excess liquid was removed with Whatman filter paper, and the grids were plunge-frozen in liquid ethane. Imaging was conducted on the Titan Krios G4 cryo-EM system equipped with a Gatan energy filter and a K3 Summit direct detector at an acceleration voltage of 300 kV. Tilt-series were acquired with *Tomography* 5 software version 5.14 (Thermo Fisher Scientific) over a tilt range of −45° to 45° (3° increments), with a cumulative electron dose of 120 e−/Å², a defocus of −8 µm, and a pixel size of 3.39 Å.

Three-dimensional reconstructions of tilt-series were performed using the *IMOD* software package[64]. After binning the tilt series to 1 k, alignment was completed using *IMOD*, followed by SIRT-based reconstruction using *Tomo3D*[65]. To enhance interpretability, the tomograms were deconvolved using *IsoNet* version 0.2[66].

### Tomogram segmentation

For the better visualisation of the tomograms, *Dragonfly* software, version 2022.2 (https://www.theobjects.com/dragonfly/index.html) was used for 3D segmentation. Tomograms were initially filtered using built-in filters to improve their clarity. Manual segmentation was conducted on 15 to 20 slices from individual tomograms, which were then used to train a neural network employing the U-Net architecture with 2.5D input dimensions (5 slices)[67]. The trained model was subsequently applied to segment the full tomograms, with manual corrections implemented as required. 2D images and 3D movies were generated using the built-in functions of the Dragonfly software.

### Negative stain transmission electron microscopy

All samples (4 µl) were prepared on Formvar/Carbon 200 mesh copper grids (Ted Pella, Inc.), glow-discharged with either a Tergeo Plasma Cleaner (Pie Scientific LLC) using water vapour or GLOQUBEplus (Quorum) using air.

Samples of the extracted Csu pili taken from the upper liquid phase (see above) were, in some cases, diluted 1:10 with Milli-Q water, depending on the pilus concentration. Stacks at the bottom of the tube were partially resuspended with a pipette tip before pipetting onto the grid undiluted. The grids were incubated for 1 min for diluted samples and 2 min for undiluted samples. Grids with undiluted samples were washed with 4 μl of Milli-Q water for 30 s to remove salts. For bacterial liquid cultures, a 1:5 dilution with Milli-Q water was prepared immediately before applying the samples to the grids, followed by a 2-min incubation. For the initial colony samples, a piece of a colony measuring 1.0 to 1.5 mm in width was picked using a 1 μl loop and suspended in 20 μl of Milli-Q water by spinning the loop for 1–2 min. For later samples, including all the *A. baumannii* AB5075[68] samples, the spinning time was only 15–20 s. When assessing for the presence of PNAG in *A. baumannii* colonies, undiluted *Triticum vulgare* lectin (WGA), colloidal gold conjugate (15 nm, CD BioGlyco) was used for suspension instead of Milli-Q water. The samples were then added to the grids and incubated for 2 min. When testing Csu pilus stack stability in the absence of PNAG and eDNA, colonies of *A. baumannii* AB5075 *pgaC:T26* mutant lacking PNAG were incubated for 1 h in 20 μl of 25 mM Tris-HCl, 6.25 mM $CaCl_2$, 1 mM $MgSO_4$, pH 7.5 with added 1 mg/ml DNase I grade II from bovine pancreas (04716728001, Roche) and 5 μM chloroquine (C6628-25G, Sigma-Aldrich). Chloroquine had the purpose of changing eDNA in Z-DNA conformation into B-DNA that DNase I can break. To reduce DNase I concentration after the reaction, each reaction mixture was diluted with 20 μl of Milli-Q water, then added to the grids in 4 μl volume and incubated for 2 min, before washing the grid for 30 s with 4 μl of Milli-Q water.

All samples were stained with 4 μl of 1% (w/v) uranyl acetate for 1 min. Images were recorded on a 120 kV JEM−1400 Plus transmission electron microscope (JEOL Ltd.) equipped with an OSIS Quemesa 11 Mpix bottom-mounted digital camera.

### Scanning electron microscopy analysis of *A. baumannii* 3D biofilms

*Acinetobacter baumannii* clinical isolates expressing Csu pili involved in biofilm formation (*A. baumannii* A100[35], *A. baumannii* AB5075[68], *A. baumannii* ATCC 19606T)[53] were cultured on LB agar at 30 °C with duration as mentioned in the respective figure legends. Pieces of agar containing bacterial colonies were sampled and fixed with 2.5% glutaraldehyde in 0.1 M sodium cacodylate overnight at 4 °C. The specimens were subsequently dehydrated in a graded series of ethanol treatment and coated with 5 nm gold/palladium as described earlier[35]. Images of the bacterial cells were obtained with a field-emission scanning electron microscope (Carl Zeiss Merlin FESEM) using secondary electron detectors at an accelerating voltage of 4 kV and probe current of 50–100 pA.

### Phylogenetic analysis of CsuA/B subunits from different *Acinetobacter* species

For the analysis, we used 29 sequences of the CsuA/B subunit from various *Acinetobacter* species, along with nine homologous sequences from different genera as an outgroup. Secretory signal peptides predicted using the SignalP 5.0 server[69] were removed. The sequences were aligned using the MUSCLE algorithm in *MEGA* 11[70], with parameters set to a −3.50 gap open penalty, −0.30 gap extend penalty, 1.50 hydrophobicity multiplier, and 18 maximum iterations. These settings effectively aligned the invariant disulphide pair and the other highly conserved residues and hydrophobic sites.

We constructed a maximum likelihood phylogenetic tree using *MEGA* 11, applying the Whelan and Goldman + Freq. model[71]. The initial tree was built by employing the Neighbour-Joining and BioNJ algorithms on a matrix of pairwise distances derived from the JTT model[71], selecting the topology with the highest log-likelihood value. To model evolutionary rate differences, we used five discrete Gamma categories, with some sites treated as evolutionarily invariable. The resulting tree features branches supported by over 50% of the 3000 replicates. For further details, see Supplementary Figs. 9a, 10.

### Reporting summary

Further information on research design is available in the Nature Portfolio Reporting Summary linked to this article.

## Data availability

The coordinates were deposited at the Protein Data Bank (PDB) with the accession codes 9I37, 9I3M, 9I3N, and 9I3O for the native Csu pilus rod, pilus pair, and type 1 and type 2 stacks, respectively. The corresponding cryo-EM maps were deposited at the EMDB with accession codes EMD-52587, EMD-52600, EMD-52601, and EMD-52602. Structure of a single Csu pilus rod, purified from recombinant *E. coli*, with PDB accession code 7ZL4 and EMDB accession code EMD-14777, was used for model building and structural comparison. We also compared our structures to the structure of a single CupE pilus rod with PDB accession code 8CIO and EMDB accession code EMD-16683. All plasmids for recombinant protein production used in this study are available upon request. A Source Data file for pilus overlap analysis is provided with this paper. Source data are provided with this paper.

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

## Acknowledgements

We thank the staff of the Cryo-EM Swedish National Facility at SciLifeLab, Stockholm, the Biochemical Imaging Centre Umeå (BICU), Umeå University and Ian Holmes Imaging Centre (Bio21, University of Melbourne) for their assistance during data collection. This work was supported by grants from the Academy of Finland (321762 and 360760) and S. Juselius Foundation (2023) to A.V.Z.; NHMRC (APP1196924) and The Cumming Global Centre for Pandemic Therapeutics Foundation to D.G.; the Swedish Research Council (SRC) (2019-01720), Kempestiftelserna (SMK21-0076) and The Faculty of Medicine, Umeå University (Insamlingsstiftelsen grant 2021-2023) to B.E.U.; SRC (2020-06136) to I.A.; and an Instrumentarium Science Foundation and a Finnish Cultural Foundation stipends to H.M.

## Author contributions

H.M. purified pili and pilus stacks and performed negative stain TEM. M.T. produced genetic constructs and performed expression tests. H.M., N.P., and A.V.Z. collected cryo-EM data and obtained cryo-EM maps. H.M. built the models, refined the structures, and performed phylogenetic analysis. B.P. and D.G. performed the cryo-ET study. I.A. and B.E.U. performed the SEM experiments. A.V.Z., D.G. and B.E.U. supervised the work. A.V.Z., H.M., N.P., B.P., D.G., B.E.U. and S.D.K. analysed the data and wrote the paper.

## Competing interests

The authors declare no competing interests.
