## [Transparent Peer Review file · Nature Communications]

Antiparallel stacking of Csu pili drives *Acinetobacter baumannii* 3D biofilm assembly

Corresponding Author: Professor Anton Zavialov

Version 0:

Reviewer comments:

Reviewer #1

(Remarks to the Author)

Malmi et al. report a novel mechanism contributing to biofilm formation in *Acinetobacter baumannii*, a common pathogen in developing countries associated with various human infections, including pneumonia as well as skin and urinary tract infections. *A. baumannii* relies on its abundant Csu pili for adherence to epithelial cells and subsequent biofilm formation, making them a major virulence factor in disease progression. While the structure of Csu pili has been recently determined—revealing a highly elastic, zig-zagged assembly with regularly spaced clinches of unknown biological significance—the precise mechanism by which they promote biofilm formation has remained unclear.

In this study, Malmi et al. use a combination of scanning and transmission electron microscopy to propose that antiparallel stacking between adjacent Csu pili is the primary driver of biofilm formation in *A. baumannii*. Although the study lacks in vivo data to support this hypothesis, the authors employ state-of-the-art cryo-electron tomography and microscopy to provide structural evidence of stacking in in vitro purified Csu pili samples following prolonged incubation at low temperatures. Their findings suggest that the clinches identified in the atomic structure of Csu pili play a critical role in the reported stacking. Overall, while the structural data strongly suggest the proposed stacking mechanism, without appropriate in vivo validation, this conclusion cannot be fully substantiated. Therefore, in its current form, the manuscript is not yet suitable for publication. If in vivo evidence supporting the structural findings were provided, the study would represent a significant and novel contribution to understanding biofilm formation and pathogenesis in *A. baumannii*. Under those conditions, I would recommend the manuscript for publication given its potentially high impact and relevance.

Major Comments:

Firstly, the absence of mutational validation is particularly striking given that Figure 6 identifies specific residues involved in stacking. This validation is essential to confirm the role of Csu pili stacking in *A. baumannii* biofilm formation and to disentangle its contribution from other polar effects, such as PNAG or eDNA accumulation. Mutants should be analysed via electron microscopy alongside biofilm formation assays to determine the precise impact of stacking. Furthermore, since the authors propose a similar mechanism for CupE pili, a comparable experimental approach would strengthen the claim that stacking is a widespread feature of archaic chaperone-usher pili, thereby enhancing the study's significance. Moreover, due to the high impact and novelty of the findings, I encourage the authors to consider fluorescence imaging of the pili to show the establishment and progression of the Csu stacking.

Secondly, the authors' claim that stacking is the 'key' mechanism of biofilm formation is not fully supported, as the extent of pili-pili interactions remains unclear. While the authors demonstrate that stacking occurs significantly in vitro under prolonged low-temperature incubation, the SEM images (Fig. 1) suggest that stacking contacts are limited to short stretches rather than spanning the entire pilus length. Consequently, it is uncertain how much these interactions contribute to in vivo biofilm formation. As the authors suggest, the presence of PNAG and eDNA likely reinforces these interactions, the primary role of Csu pili in biofilm formation could be solely to sterically trap large amounts of PNAG and eDNA rather than stacking with itself. To clarify this, the authors should generate a non-stackable Csu mutant (as mentioned previously) to determine whether stacking is the 'key' or rather a 'supporting' mechanism. Another approach would be to assess biofilm formation in the presence and absence of PNAG/eDNA using enzymatic digestion. This would allow for the isolation of pili-pili interactions, helping to determine whether stacking is indeed the key mechanism or a secondary effect.

Specific comments

L71-73: The authors state that the measured thickness of Csu pili is ~30 nm (Fig. 1). However, it is evident from the image

that this thickness likely results from the bundling of filaments rather than individual pili. Could the authors clarify how these measurements were obtained? How many bundles were considered? Was there a range of values? Presenting this data as mean \pm SD would enhance transparency, particularly given that this observation prompted the study.

L76: The authors base their in vivo observations on a translucent phenotype, which is generally associated with increased pilus production. To ensure that the observed stacking is not an artefact of higher pili numbers, could the authors provide similar images to those in Figure 2 for a wild-type *A. baumannii* strain?

L77-81: Authors show that stacks are not formed effectively in liquid cultures and are present in colonies. Would that not indicate that it is rather the accumulation of ECM (possible only under colony growth and not in liquids) that is primarily responsible for pili bundling and cell aggregation rather than the inherent pili-pili interaction which should be capable of forming even in liquid environments?

L82-87: In the cryo-ET results (Fig. 3), there appears to be no continuity between pili from neighbouring cells to confirm that stacks are intercellular rather than intracellular, suggesting that the shown stacks originate from the same cell. Given the claims of exclusively anti-parallel stacking, how do the authors reconcile this observation?

L99-101: The authors state that a CsuE mutant with a compromised tip finger site impairs adhesion but does not affect Csu pili stacking, and consequently, biofilm formation remains unaffected. This statement requires quantification—biofilm formation levels should be explicitly compared to support the claim of unaltered levels.

L112-115: The progressing nature of assembly has been only shown at low temperatures in vitro and therefore should not be mentioned confidently unless authors could show it through in vivo imaging. Moreover, the assertion that the biofilm-like structures observed in vitro indicate that biofilm formation is a “spontaneous process inherently programmed within the structure of the pili” is strong but lacks consideration of alternative explanations. Could environmental factors, such as salinity or pH, also contribute to stack formation? Testing whether stacks still form under high ionic strength conditions would help exclude non-specific interactions driven by low ionic strength. Additionally, is stacking reversible under certain environmental conditions?

L135-141: During grid preparation and freezing, did the authors observe any dependence of stacking on ice thickness? Could the thin ice required for SPA induce or perhaps also limit the formation of Csu pili stacks?

L189-202: I strongly encourage the authors to provide mutational validation for their observations regarding CupE pili. This would significantly strengthen the novelty of their claims, allowing for extrapolation of their findings to all archaic pili based on experimental data, increasing impact.

L204: The claim that PNAG embeds pilus stacks in mature biofilms is based on WGA labelling. To confirm this, could the authors treat samples with PNAG-hydrolyzing enzymes and analyze the effect via TEM imaging? Additionally, how do biofilm levels compare in the presence and absence of this enzyme? Could authors discuss the possibility that trapping PNAG and eDNA is the main involvement of Csu pili in biofilm formation rather than a side effect of stacking.

L237-240: Are there any documented differences in the production of Csu pili in liquid vs biofilm growth that could support this?

L251-252: Has it been tested—either in this study or elsewhere—whether non-zigzagged pili are unable to form gel-like structures after prolonged incubation at low temperatures? If not, this statement should be softened.

L269-270: Structurally, if one site on the pilus rod is occupied by another to form a stack, why would the remaining binding sites become inaccessible for another junction? Can the authors confidently exclude the possibility of three-dimensional stacking instead of exclusively flat stacking? Is there any in vivo evidence supporting the formation of flat connections rather than thick, circular bundles? If flat stacks were the predominant arrangement, wouldn't they be visible in SEM images? While the cryo-ET data suggests flat stacks, without subtomogram averaging, this conclusion remains uncertain.

L233-236: The authors suggest that increased turbulence explains why liquid cultures do not exhibit the same levels of Csu-mediated biofilm formation. Could they provide evidence demonstrating the impact of growth conditions on Csu stack formation in vitro? Additionally, could they visualize the effect of turbulence on pre-assembled Csu stacks?

Reviewer #2

(Remarks to the Author)

Reviewer #3

(Remarks to the Author)

I. Overall review:

The archaic chaperone usher Csu pili formed by *Acinetobacter baumannii* are known to play essential roles in the formation of biofilms. Csu pili have been previously shown to biofilms on abiotic surfaces as well as biofilm pellicles and play crucial roles in adhesion to mammalian tissues. This study by Malmi et. al uses a variety of electron microscopy techniques (SEM, Negative stain TEM, Cryo-ET, Cryo-EM, and gold conjugation labeling) to investigate the role that Csu pili play in biofilm formation at multiple length scales from the micro/nanoscale of SEM and cryo-ET to sub-nanometer and atomistic details provided by single particle cryo-EM. Using these length scales the authors are able to strongly suggest how the Csu pili are involved in biofilm formation. The cryo-EM structural details of the pilus pairs and stacks, while low-resolution, are able to show novel anti-parallel packing interactions between pili. When attached to cells these anti-parallel interactions would presumably mostly occur between different cells which is plausible. Upon examination of the atomic models and cryo-EM density maps, I am confident in the authors' interpretation of most of their cryo-EM structures. Any concerns are noted in the

next section of my review. With all of this data the authors propose a plausible model for Csu pili-mediated biofilm formation.

Overall, this study is rigorous, well-written, well-presented, and should have appeal to a broad scientific readership. Upon sufficient response to my concerns from the authors, I think this paper would be an excellent fit for publication at Nature Communications. I have several points which I would strongly like to see the authors address as well as some minor suggestions that could help improve their story/results.

II. Major points to be addressed:

1. In the Methods on page 16 lines 30-31 it is stated: "Homogeneous Refinement using our earlier 8.5 Å map (unpublished) of type 1 stack revealed that there are two types of stacks". This seems very unlikely that two types of stacks were revealed this way. Do the authors mean "Heterogenous Refinement"?
2. For the Csu pili pairs, the results section on page 5 lines 30-32 state that: "Since the pilus pair lacks helical symmetry, we generated an initial model based on the structure of the single pilus rod by optimizing the overlap between its projections and a 2D class average (Supplementary Fig. 10–12)". This model was built prior to reconstruction of the pilus pairs map and then docked into the map. Do the authors find any significant differences if they build the model by docking individual subunits into the low-resolution map?
3. Map:Map FSC curves for each of the structures from cryoSPARC should be presented as supplementary figures with resolution and the either the 0.143 or 0.5 FSC line being indicated.
4. Workflows for each cryo-EM structure should be presented so readers can easily follow how the maps were generated. These should start with particle picking and show each step in the reconstruction process (heterogenous refinement, homogenous refinement, post processing, .etc).
5. The model in Figure 7 proposed for Csu pili-mediated biofilm formation focuses on cell division as the means from which more cells are introduced into this biofilm. *Acinetobacter baumannii* have been shown to have T4P mediated twitching motility. Could this also significantly contribute to biofilm formation? If so, such considerations should be introduced into the paper. If not, I would suggest that the authors also address this briefly in the manuscript discussion.
6. Building off of this, type IV pili appear to not be mentioned at all in the paper despite their presence in this strain of *Acinetobacter baumannii* (I believe). Were any T4P detected during cryo-EM image processing (2D classification steps, .etc)? Given the robust results, I have little doubt that the stacks/networks are formed by the Csu pili, I am just curious whether any T4P were detected at all.
7. In figures 2-3 the pili stacks/bundles are quite hard to see in the negative stain TEMs (Fig. 2d-f mostly) as well as in the tomogram slices (Fig. 3a-c). To an untrained eye, which would include much of the broad readership of Nature communications, some of these panels might difficult to interpret even with the helpful annotations. Is there any way that higher contrast images can be used for some of these panels? For figure 2, I have absolutely no idea how to interpret panel 2f. I can detect no difference between stacks and networks in that panel despite the presence of the diamonds. For figure 3 a-c it might be possible to average together several adjacent Z-slices to generate better contrast. Would this be feasible?

III. Additional suggestions which may improve the story/results

1. Supplementary Fig. 7 might be better as a main figure as its results are key to the model proposed in current main Figure 7.
2. The number of particles used to obtain the type 1 and type 2 pili stacks was quite a lot given the relatively low resolutions achieved. Perhaps heterogeneity is the limiting factor in terms of resolution. I wonder if more classes of pili stacks would be revealed with additional heterogeneity analyses such as cryoSPARC's 3D variability analysis (cluster output might be most useful) or cryoDRGN?

Version 1:

Reviewer comments:

Reviewer #1

(Remarks to the Author)

The authors have successfully implemented all necessary changes and improvements, resulting in a significantly better and more substantial paper. I recommend its publication in its current form.

Reviewer #2

(Remarks to the Author)

I co-reviewed this manuscript with one of the reviewers who provided the listed reports. This is part of the Nature Communications initiative to facilitate training in peer review and to provide appropriate recognition for Early Career

Researchers who co-review manuscripts.

Reviewer #3

(Remarks to the Author)

The authors have more than addressed all of my concerns. I view this work as impactful and within the scope of Nature Communications and I recommend this manuscript for publication. From a cryo-EM perspective the reconstruction of the different classes of pili stacks was technically challenging and impressive, despite the fact that the authors only achieved low to modest resolution of the stacks. Their high-resolution structure of the pilus rod as well as the fact that they use isolated stacks from a Csu E. coli expression system leave little ambiguities to the sequence of the pilin subunit in these structures. I have several minor suggestions that I think would improve the manuscript.

1. In Figure 5A the authors should include text displaying the resolution of each structure shown.
2. The authors stated in their rebuttal "To obtain most of our SPA cryo-EM structures, including the Csu pilus stacks, we expressed Csu pili using our E. coli BL21 pBAD-Csu overexpression system". It should be made very clear in the methods which expressed systems were used for which structures.

Detailed responses to the reviewers' comments

Responses to reviewer 1, pages 1–19

Responses to reviewers 2 and 3, pages 20–32

References, pages 32–34

Reviewer 1

We were pleased to read that the reviewer found that our work is has “potentially high impact and relevance” and “would represent a significant and novel contribution to understanding biofilm formation and pathogenesis in *A. baumannii*”. We thank the reviewer for a thorough assessment of our manuscript and helpful suggestions.

General points:

Overall, while the structural data strongly suggest the proposed stacking mechanism, without appropriate in vivo validation, this conclusion cannot be fully substantiated. Therefore, in its current form, the manuscript is not yet suitable for publication. If in vivo evidence supporting the structural findings were provided, the study would represent a significant and novel contribution to understanding biofilm formation and pathogenesis in *A. baumannii*. Under those conditions, I would recommend the manuscript for publication given its potentially high impact and relevance.

We initially chose not to include additional *in vivo* validation because the critical role of Csu pili in *A. baumannii*, as well as CupE pili in *P. aeruginosa*, in 3D biofilm formation has already been demonstrated in three independent studies, as summarized in **Box 1**. We apologize for not stating this more clearly in the Introduction.

Although the mechanism of archaic CUP pilus-mediated 3D biofilm formation was previously not known, these studies firmly established the importance of this phenomenon and explored its relationship to PNAG and eDNA—addressing several points raised by the reviewer.

Box 1. Summary of *in vivo* evidence from previously published studies.

(**Romero et al, 2022**) demonstrated that deletion of the Csu pilus assembly platform (CsuD usher) in *A. baumannii* strain ATCC 17978 abolishes the formation of mushroom-shaped 3D biofilms (see Fig. 2 in their paper). Given that this study observed the natural accumulation of *A. baumannii* cells from a liquid culture onto a glass plate and the subsequent biofilm development over four days, it offers strong *in vivo* evidence.

(**Ahmad et al, 2023**) covered this topic even more thoroughly through various deletion mutants. As can be seen in their Fig. 2A–B, the deletion of Csu pilus neck-subunit CsuA in the strain ATCC 17978 turns the normally thick patches of cells on glass plates into flat monolayers, which is also apparent in the relative OD595 and measured biomass. This also directly correlates with the presence of Csu pili on the bacterial surface, as they show in their Supplementary Fig. 2A. Note that for the Fig. 2A, they also tested a deletion of the chaperone CsuC, which also collapsed biofilm mass, and restoration of CsuA through a plasmid, which brought biofilm mass back to wild type levels. Furthermore, as shown by their Fig. 2A–B and Supplementary Fig. 2A–C, their deletion of Type IV pilus main subunit PilA causes an increase in CsuA/B expression, more Csu pili on the surface, as well as thicker biofilms with higher biomass. Finally, they also repeated these results in the hyper-virulent strain AB5075, whose wild type form produces an abundance of Csu pili, as shown by their AFM image in Fig. 3C. Their Fig. 3F and Supplementary Fig. 2D–E show that transposon insertions into the genes for Csu pilus main subunit CsuA/B and the PilA in strain AB5075 matched the effects of the equivalent deletions in the strain ATCC 17978. The authors also observed accumulation of PNAG in mountain-like biofilm structures (Fig 1D in the paper); however, their mutagenesis data clearly demonstrate the dominant role of Csu pili in 3D biofilm formation.

Similarly, by deleting genes in the cluster encoding CupE pili in *P. aeruginosa*, (**Giraud et al, 2011**) demonstrated the key role of CupE pili in cohesive biofilm structuring: the absence of CupE pili abolished the formation of three-dimensional, mushroom-shaped bacterial assemblies (see Fig. 6 in their paper). Like Csu pili in *A. baumannii*, CupE pili in *P. aeruginosa* are assembled via the archaic CUP, suggesting a broader, possibly conserved role for archaic CUP systems in 3D biofilm formation.

To explain these important prior findings better, we have added the following sentences (highlighted with yellow background shading here and in subsequent sections) to the Introduction (page 3):

“Classical and alternative CUPs primarily facilitate initial, highly specific adhesion. In contrast, archaic CUP systems are associated with the establishment of mature 3D biofilms. (Ahmad *et al.*, 2023; Ching *et al.*, 2024; Giraud *et al.*, 2011; Romero *et al.*, 2022) Among the most troublesome Gram-negative bacteria, *Acinetobacter baumannii* and *Pseudomonas aeruginosa* (Cerqueira & Peleg, 2011; Tuon *et al.*, 2022) rely on archaic Csu and CupE pili, respectively, to develop stress and drug-resistant mushroom or mountain-like 3D biofilms on a diverse array of biotic and abiotic surfaces (Ahmad *et al.*, 2023; Ching *et al.*, 2024; Giraud *et al.*, 2011; Romero *et al.*, 2022; Tomaras *et al.*, 2003). Although the central role of Csu and CupE pili in 3D biofilm formation has been firmly established through deleterious mutagenesis of structural or assembly genes from both archaic systems (Ahmad *et al.*, 2023; Giraud *et al.*, 2011; Romero *et al.*, 2022), the underlying mechanism of this process remained elusive.”

Major Comments:

(1.1) Firstly, the absence of mutational validation is particularly striking given that Figure 6 identifies specific residues involved in stacking. This validation is essential to confirm the role of Csu pili stacking in *A. baumannii* biofilm formation and to disentangle its contribution from other polar effects, such as PNAG or eDNA accumulation. Mutants should be analysed via electron microscopy alongside biofilm formation assays to determine the precise impact of stacking.

The importance of Csu pili in establishing 3D biofilms has already been extensively validated *in vivo* by two independent studies (see our detailed response to General points above and **Box 1**). These studies firmly establish that Csu pili are critical to 3D biofilm formation and explore their relationship with PNAG and eDNA. However, we agree with the reviewer that mutagenesis of the stacking interfaces is valuable, as it would also provide important mechanistic insight on how the binding junction forms. In fact, we sought to identify the binding junction residues through mutagenesis before we achieved sufficient resolution in the cryo-EM maps to model the binding contacts. This approach, however, faced two major challenges.

First, designing mutants that completely disrupt binding is challenging because the junction residues are surrounded by highly conserved regions that are critical for donor strand exchange and pilus assembly (see “Supplementary Notes,” section “Antiparallel Binding Site Adjustment”). This limits the use of larger mutations, such as deletions or insertions. Conversely, introducing point mutations at the binding residues presents a different challenge: the binding junction itself appears to be tolerant, accommodating multiple residue combinations. This is supported by our phylogenetic analysis of CsuA/B from different species (current Supplementary Fig. 9a and 10), which predicts considerable sequence variability at these positions. Our modelling (Supplementary Fig. 9b–d) also indicates that these variants should work similarly well requiring no major changes to subunit orientations or structure.

Second, the high heterogeneity of stack networks limits the techniques that could be utilized to assess the effects of mutations. Each stack can include Csu pili of varying lengths, each forming a variable number of binding junctions with neighbouring pili. Additionally, the number of Csu pili per stack can vary significantly, making biophysical experiments impractical, as they cannot be reliably reproduced. The pili are also too thin to be individually visualized using fluorescence microscopy and our anti-CsuA/B polyclonal antibodies with two binding sites each would likely get in the way of stack formation, while also connecting the pili randomly. This leaves negative stain TEM and cryo-EM as the only viable methods for detecting the presence of stacks. However, these techniques cannot assess the strength of the binding interactions. Therefore, unless the mutant prevents the binding entirely, we need to look for various cues, such as the accumulation rates of pairs and stacks

compared to the number of individual pili, which is quite open to interpretation. This challenge is further compounded by variability in the rate of stack formation: while stacks form within hours-days in biofilms (*in vivo*), they may take over a month to develop in purified samples (*in vitro*).

Because of these limitations, we adopted two alternative approaches to inhibit pilus stacking. In the first approach, we introduced aspartate and glutamate residues opposite to each other in the binding junction to create electrostatic repulsion between the two surfaces. The second approach leveraged the predicted differences between the binding junctions of *A. baumannii* ATCC 19606 and *A. defluvii* WCHA30 (Supplementary Fig. 9). Specifically, *A. defluvii* WCHA30 has a two-residue deletion in the C2–D loop and a single-residue deletion in the B'–C1 loop, and modelling these into the *A. baumannii* binding junction results in an even, approximately 2 Å wide gap between the pili, which is predicted to close by the pili moving closer. We aimed to replicate this effect by introducing only one of these deletions into the binding junction, thereby inhibiting stack formation by preventing the gap from closing.

The results of this study are presented in the new Supplementary Table 3. Although most of the mutations either prevented or severely impaired pilus assembly/secretion, rendering them unsuitable for further analysis, the T66D–A89D double mutation and the same mutation with an additional deletion Δ P68 were well tolerated. Both mutations negatively affected the formation of larger stacks *in vitro*, with the effect being somewhat clearer in the triple mutant (new Supplementary Fig. 8). These results provide independent support for the proposed mechanism of pilus stacking and highlight the adaptability of binding contacts, as predicted by our phylogenetic analysis. We have added the following sentences to the main text (page 7):

“We next examined the contribution of individual residues to stack formation. Most mutations at the pilus stacking interfaces also severely disrupted pilus assembly (Supplementary Table 3). However, the double mutant T66D–A89D, designed to introduce repulsive negative charges, and its variant T66D– Δ P68–A89D, featuring a shortened B'–C1 loop to position Asp66 closer to Asp89 and to prevent Asp66 forming hydrogen bonds with the main-chain of the opposing Csu pilus (Supplementary Fig. 8a, b), were well tolerated. Although these mutations did not abolish antiparallel interactions *in vivo*, the more sensitive *in vitro* assays using purified pili showed that the mutants had smaller, less organized stacks with shorter continuous stacked regions compared to the wild type (Supplementary Fig. 8c, d), consistent with our structural data.”

(1.2) Furthermore, since the authors propose a similar mechanism for CupE pili, a comparable experimental approach would strengthen the claim that stacking is a widespread feature of archaic chaperone-usher pili, thereby enhancing the study's significance.

We fully agree with the reviewer. Unfortunately, the *P. aeruginosa* CupE system is challenging to work with. While *A. baumannii* Csu pili are expressed in large quantities across many *A. baumannii* strains, the expression of CupE pili is tightly regulated and occurs only under specific conditions that strongly favour the sessile growth of this normally flagellated bacterium. Additionally, *P. aeruginosa* expresses several other chaperone-usher pathway (CUP) systems, necessitating the use of deletion mutants to prevent confusion during analysis.

Our prediction is based on the excellent work by (Bohning *et al*, 2023), who observed a “crisscross mesh-like array” of CupE pili *in vitro* (Supplementary Figure 1D in their paper), which, in our opinion, closely resembles the Csu pilus stacks. We also clearly observe pilus pairs in their images (Figure 1A; Supplementary Figure 1B). To increase CupE expression and enable structural analysis, Böhning *et al.* engineered a triple mutant strain (Δ *pilA* Δ *fliC* Δ *mvaT*). Given that Böhning *et al.* are conducting similar studies and possess greater expertise with this organism, we plan to collaborate with their group to solve the structure of CupE pilus stacks, which we intend to publish in a separate manuscript.

To address the reviewer's concern, we have tempered our claim regarding the similar mechanism for CupE pili. Specifically, we added "potential" to the following sentence on page 8:

"Its position indicates it could be part of an antiparallel CupE pilus included in the helical reconstruction, which allowed us to model the potential antiparallel junction (Supplementary Fig. 11b)."

In the following paragraph, we also changed "it is quite likely" to "is plausible":

"Therefore, it is plausible that the archaic CU pilus zigzag architecture, which enables the formation of flat stacks, evolved to shape bacterial communities."

(1.3) Moreover, due to the high impact and novelty of the findings, I encourage the authors to consider fluorescence imaging of the pili to show the establishment and progression of the Csu stacking.

We greatly appreciate this valuable suggestion. However, Csu pili are too thin to individually visualize under confocal microscopy. Implementing such experiments would require the development of an entirely new experimental setup and this is therefore beyond the scope of the current manuscript. However, we are actively pursuing this direction in our ongoing work. Recently, we discovered that Csu pili (and likely most archaic CUP systems) form biofilms on oil surfaces due to the unique hydrophobic finger-like structure at the tip subunit, CsuE. Analysing biofilms on oils using confocal microscopy offers an excellent system to study Csu-mediated adhesion to surfaces and 3D biofilm formation (see Fig. R1 below). However, a new method is still needed to visualize pilus stacks in this context. Once the complete method is established, we plan to carry out these experiments and publish a separate paper on Csu-mediated biofilm formation on oils, as it will include a substantial amount of new data.

Fig. R1 | Csu pilus mediated *A. baumannii* biofilm formation on a drop of mineral oil. The first (0 min) and the last (2 h 40 min) frames of a video recorded on a confocal microscope.

We have added the following sentences to the main text in Discussion on page 11:

"Csu pilus stack formation is a surprisingly rapid process, occurring simultaneously with 3D biofilm growth. Our results suggest that the fast kinetics may be explained by the initiation of pilus stacking around the septa of dividing cells. Pilus-pilus contacts may form quickly through a zipper-like mechanism, in which the establishment of each binding junction facilitates the formation of the next. The high cooperativity of this process may depend on the relatively flexible tips of the B''-C1 and

D''–E1 loops (Supplementary Fig. 7c), which, according to our bioinformatic analysis and mutagenesis data, can accommodate a range of binding geometries. Further studies will be needed to clarify this mechanism.”

(2.1) Secondly, the authors’ claim that stacking is the ‘key’ mechanism of biofilm formation is not fully supported, as the extent of pili-pili interactions remains unclear. While the authors demonstrate that stacking occurs significantly *in vitro* under prolonged low-temperature incubation, the SEM images (Fig. 1) suggest that stacking contacts are limited to short stretches rather than spanning the entire pilus length. Consequently, it is uncertain how much these interactions contribute to *in vivo* biofilm formation.

Response on temperature:

Before addressing to the reviewer’s comment, we would like to clarify a major misunderstanding regarding the temperature that likely arose from our failure to specify clearly the temperatures at which Csu pilus stacks form *in vivo*. Unlike in our *in vitro* experiments, **Csu pilus stacking networks form very rapidly *in vivo* across a broad range of ambient and physiological (body) temperatures, progressing in parallel with 3D biofilm formation.** The sole purpose of storing the plates at 8°C was to prevent overgrowth of the colonies.

To address this point, we analysed stack formation in *A. baumannii* colonies immediately after growth at 22°C, 30°C, and 37°C using negative stain TEM (new Supplementary Fig. 2a–c). Extensive networks of pilus stacks were observed at all three temperatures in the hypervirulent strain AB5075 of *A. baumannii*.

Furthermore, our cryo-electron tomography experiment (Fig. 3) was conducted on an *A. baumannii* 19096T colony grown for three days at 22°C, without any low-temperature storage. Likewise, *A. baumannii* liquid cultures grown at 22°C (Supplementary Fig. 3; please, note that figure numbers have changes) and recombinant *E. coli* liquid cultures grown at 37°C were not stored prior to analysis. The same applies to all samples in our new negative stain TEM experiments (new Fig. 8 and Supplementary Fig. 1–2, 4).

We have added the following sentence to the main text on page 4:

“These structures formed equally efficiently under both ambient and physiological temperatures (22–37°C) and across a broad range of pH (6.0–8.5) and salt concentration (42.8–400 mM) (Supplementary Fig. 2).”

We have also clarified the conditions of cell growth in the Methods section and relevant figure legends.

Response on “the SEM images (Fig. 1) suggest that stacking contacts are limited to short stretches rather than spanning the entire pilus length”

We thank the reviewer for this observation. Unfortunately, SEM offers low resolution and is prone to dehydration artefacts. Moreover, most of the longer stacks in colonies are not straight but instead follow the curvature of the bacteria, akin to a sheath, which makes them particularly difficult to capture with SEM. Therefore, to address this point, we focused on the analysis of negative-stain TEM and cryo-electron tomography.

Both our negative stain TEM and cryo-electron tomography *in vivo* experiments reveal highly extensive networks of Csu pili, with stacks forming extremely long and continuous contacts. In our

new Supplementary Figure 1b, we highlight a traced 7920 Å section of a Csu pilus within a stack, showing that ~6980 Å of this section is engaged in interactions—roughly half the pilus length. This correlates to around 42 binding junctions with a combined interactive surface of ~17,470 Å². Moreover, stacks formed *in vivo* can be rather wide, with the one we observed in Supplementary Fig. 1c consisting of 15 adjacent Csu pili. Finally, the local Csu pilus stacks may also connect with one another to form superstacks, linear regions of stacked pili far exceeding the length of individual Csu pili (>12 µm vs. 1.7 µm) by continuously receiving new pili from the bacterial surfaces at regular intervals, wrapping around from multiple bacteria (see Fig. 2f; Supplementary Figure 1d–h).

- Please note that the resolution of the TEM figures in the automatically generated pdf file is not optimal. The images are better seen in the docx source files at a 125 % zoom.

We have inserted the following sentences into the main text page 4:

“Stacks may contain more than 15 pili, with pili often forming lateral interactions extending over half of their length (Supplementary Fig. 1a–c). Superstacks, continually incorporating new pili from the surfaces of multiple bacteria, can exceed 12 µm in length (Supplementary Fig. 1d–h), clumping bacteria in 3D structures (Supplementary Fig. 2). These structures formed equally efficiently under both ambient and physiological temperatures (22–37°C) and across a broad range of pH (6.0–8.5) and salt concentration (42.8–400 mM) (Supplementary Fig. 2).”

and page 8:

“Such unusually short repeats are possible only because of the zigzag architecture of the pilus, which generates a high density of binding junctions. For example, in a typical stack observed *in vivo*, a traced Csu pilus ran ~792 nm antiparallel to adjacent pili forming ~42 binding junctions with a predicted total interactive surface of around 17470 Å² (Supplementary Figure 1b). Therefore, it is plausible that archaic CU pilus zigzag architecture, enabling the formation of flat stacks, evolved to shape bacterial communities.”

(2.2) As the authors suggest, the presence of PNAG and eDNA likely reinforces these interactions, the primary role of Csu pili in biofilm formation could be solely to sterically trap large amounts of PNAG and eDNA rather than stacking with itself. To clarify this, the authors should generate a non-stackable Csu mutant (as mentioned previously) to determine whether stacking is the ‘key’ or rather a ‘supporting’ mechanism. Another approach would be to assess biofilm formation in the presence and absence of PNAG/eDNA using enzymatic digestion. This would allow for the isolation of pili-pili interactions, helping to determine whether stacking is indeed the key mechanism or a secondary effect.

We thank the reviewer for this insightful suggestion. While previous *in vivo* studies have established that Csu pili are essential for linking bacterial cells in 3D biofilms (see response above and **Box 1**), they did not conclusively demonstrate that this occurs independently of PNAG and eDNA. As the reviewer pointed out, pili could instead facilitate the accumulation of PNAG and eDNA, which would then glue the bacteria together. Resolving this issue requires direct proof that Csu pili and their stacks alone can hold bacteria together in the absence of PNAG and eDNA. We agree that non-stackable mutants would provide particularly direct evidence. However, due to the technical challenges of generating mutants that completely abolish stacking without impairing pilus assembly or secretion (see response to Main Comment 1), we were unable to pursue this approach.

Instead, we implemented the reviewer’s alternative suggestion by assessing biofilm formation in the absence of PNAG and eDNA. Using an *A. baumannii* transposon mutant (*pgaC::T26*) lacking PNAG and enzymatic treatment with DNase I (with chloroquine to enhance eDNA degradation), we observed that bacterial clumps and rafts still formed extensive networks of stacked Csu pili even in

the complete absence of PNAG and eDNA (new Fig. 8). These findings demonstrate that Csu pili stacking alone provides strong cohesion and structural integrity, while PNAG and eDNA serve only as additional, non-essential reinforcements.

Together, these results support the conclusion that stacking is not a secondary effect, but rather the primary mechanism by which Csu pili mediate bacterial linkage and 3D biofilm formation.

We have inserted the following sentences into the main text on page 9:

“While most bacteria seen in our negative stain TEM samples formed dense clumps with aplenty of Csu pilus stacks on the surfaces and PNAG or eDNA filling the tight gaps (Fig. 8a–c, Supplementary Fig. 2), others formed loose, yet cohesive pilus-linked rafts with otherwise empty, wide gaps (Fig. 8d–f). The distinctive lack of PNAG and eDNA in these rafts indicates that these EPS contribute little to clump cohesion. To test this further, we grew *A. baumannii* AB5075 mutant *pgaC::T26*, which is unable to produce PNAG (Gallagher *et al*, 2015), and treated suspended colonies with DNase I and the DNA-intercalating agent chloroquine to degrade eDNA (Buzzo *et al*, 2021). Subsequent dilution and washing steps revealed several highly organized rafts, with tight, clear gaps and extensive Csu pilus stacks traversing them (Fig. 8g–i), further supporting the conclusion that Csu pilus stacks link the bacteria in early 3D biofilms independently of PNAG and eDNA.”

At the same time, we cannot exclude a possible role of other EPS at different stages of *A. baumannii* 3D biofilm formation. Therefore, we added the following text in Discussion (page 11):

“Our results support the dominant role of archaic Csu pili in clumping bacteria into 3D structures. However, a role for other EPS, such as PNAG and eDNA, at different stages of this process cannot be excluded. Similarly, several protein surface receptors, including OmpA (Gaddy *et al*, 2009), Ata (Bentancor *et al*, 2012) as well as Biofilm-Associated Protein (Bap) and its closely related Blp1 and Blp2 (Brossard & Campagnari, 2012; De Gregorio *et al*, 2015) facilitate biofilm formation. As these receptors have smaller structures, they likely mediate more localized interactions compared to the Csu pili. In addition, *A. baumannii* produces Type IV pili, and its different variants promote motility in AB5075 strain and biofilm formation in ACICU strain (Ronish *et al*, 2019). Interestingly, the latter strain has a transposase gene inserted between *csuA/B* and *csuA* within its *csu* operon. In our colonies of *A. baumannii* 19096T and AB5075, we observed little to no filaments wide enough (72 Å) (Wilharm *et al*, 2013) to be interpreted as Type IV pili. Only colonies grown at 30 °C and 37 °C displayed a few bacteria with such filaments, while most cells expressed abundant Csu pili. This agrees with the observation that cyclic AMP acts as a master regulator, suppressing the expression of Csu pilus subunits, PNAG-synthesizing enzymes and quorum sensing, and thus biofilm formation in general, while strongly upregulating Type IV pilus subunits and other motility-related genes (Harkova *et al*, 2024). Therefore, while Type IV pili may contribute to biofilm formation by mediating adhesion to certain surfaces, they likely lose relevance once *A. baumannii* switches to producing Csu pili. However, the expression regulation could be different in other species. Studies of other archaic CUP pili, such as the *P. aeruginosa* CupE system, may help clarify this.”

Specific comments

(3) L71-73: The authors state that the measured thickness of Csu pili is ~30 nm (Fig. 1). However, it is evident from the image that this thickness likely results from the bundling of filaments rather than individual pili. Could the authors clarify how these measurements were obtained? How many bundles were considered? Was there a range of values? Presenting this data as mean ± SD would enhance transparency, particularly given that this observation prompted the study.

We thank the reviewer for this point. In our original sentence, “*However, our analysis shows the measured thickness of these pili is ~30 nm (Fig. 1), which is 10-fold wider than a single Csu pilus rod filament,*” we intended to emphasize that the thickness of “pili” visible in the low-resolution SEM images used in several previous publications does not correspond to the actual thickness of a single pilus as determined by cryo-EM, and therefore must represent a bundle (stack). The ~30 nm fragment shown in Fig. 1 was included only as an illustrative example. We now realize that this sentence was indeed confusing, and in the revised manuscript we have clarified the text on page 4 as follows:

“Earlier studies of *A. baumannii* biofilms using scanning electron microscopy (SEM) have shown that bacterial cells are interconnected by Csu pili (Ahmad *et al.*, 2019; Tomaras *et al.*, 2003). However, our analysis of SEM images indicates that the thickness of these ‘pili’ is often much greater than the width of a single Csu pilus rod filament determined by cryo-EM (Pakharukova *et al.*, 2022). In the example shown in Figure 1, the apparent width is ~30 nm, which is 10-fold greater than that of a single pilus. This prompted us to examine the Csu pili on bacteria grown in liquid and on solid media at higher resolution using negative staining transmission electron microscopy (TEM) (Fig. 2).”

For quantitative analysis of *in vivo* stacks, we rely on negative stain TEM and cryo-ET, which preserve the native state and allow us to assess stack organization in detail. These reveal that stack width varies considerably with different numbers of Csu pili being included in it at different positions in the stacking network, making a mean \pm SD for a group of stacks misleading. Representative measurements *in vivo* revealed a stack of 15 pili with the width of 72.7 ± 1.6 nm ($n=5$) (new Supplementary Fig. 1c), which most closely correlates to a type 1 stack. Thus, the SEM observation is consistent with our higher-resolution analyses and the measured ~30 nm thickness reflects the presence of stacked pili.

We have inserted the following sentences into the main text page 4:

“Stacks may contain more than 15 pili, with pili often forming lateral interactions extending over half of their length (Supplementary Fig. 1a–c). Superstacks, continually incorporating new pili from the surfaces of multiple bacteria, can exceed 12 μ m in length (Supplementary Fig. 1d–h), clumping bacteria in 3D structures (Supplementary Fig. 2). These structures formed equally efficiently under both ambient and physiological temperatures (22–37°C) and across a broad range of pH (6.0–8.5) and salt concentration (42.8–400 mM) (Supplementary Fig. 2).”

We also mentioned the width of the 15-pilus stack (72.7 ± 1.6 nm, $n=5$) in the legend to Supplementary Fig. 1c.

(4) L76: The authors base their *in vivo* observations on a translucent phenotype, which is generally associated with increased pilus production. To ensure that the observed stacking is not an artefact of higher pili numbers, could the authors provide similar images to those in Figure 2 for a wild-type *A. baumannii* strain?

The opaque and translucent phenotypes of *A. baumannii* are both considered wild type, as there is no underlying genetic difference between them—only variations in expression levels regulated by several transcription factors (Tipton *et al.*, 2015). For our purposes, the key distinction is that the translucent phenotype produces an abundance of Csu pili (Ahmad *et al.*, 2019), promoting biofilm formation (the native sessile form), whereas the opaque phenotype predominantly expresses Type IV pili, which facilitate motility of individual cells (the native motile form). A recent article by (Harkova *et al.*, 2024) revealed that cyclic AMP (cAMP) most likely controls this switch in gene expression.

While this phenotypic difference may seem ideal for our experiments, the high rate of switching between the two states poses a challenge. Moreover, even opaque colonies retain some expression of Csu pili (Ahmad *et al.*, 2019), possibly due to the constant phenotype switching of individual

bacteria (Tipton *et al.*, 2015), which poses a challenge for a reproducible analysis if not considered while picking bacteria from colonies for the experiments.

We initially selected for the translucent phenotype following the protocol of (Anderson & Rather, 2019) due to two reasons: (1) to ensure efficient yield of Csu pili from *A. baumannii* 19096 grown in a liquid culture to facilitate structural determination of a single Csu pilus from *A. baumannii* (Fig. 5) and (2) to avoid inconsistency between experiments that may occur due to colony phase variation switch. The selection process is summarized in **Box 2**.

Box 2. Selection of translucent phenotypes of *A. baumannii*

1. Pick a colony representing a chosen phenotype (thus most cells belong to that phenotype), suspend it in medium, and grow new colonies so small that it is unlikely that the bacteria in them have switched their phenotype yet. The phenotype is not yet apparent in these barely visible colonies.
2. Grow several 5 ml liquid cultures, each using one of the tiny colonies, but stop their growth before OD595 reaches 0.1, so the bacteria are less likely to have switched their phenotype yet.
3. Make glycerol stocks of each liquid culture. Later, check by plating on Petri-dishes that most colonies formed from a glycerol stock match the desired phenotype. Keep only the good stocks.

The instructions provided very specific guidance to ensure that the bacteria would not have an opportunity to switch phenotypes. However, our examination of fully-grown colonies from both phenotypes revealed that many bacteria had already undergone phenotypic switching. Negative stain TEM of opaque colonies showed bacteria forming large, tight clumps as well as loose rafts with Csu pilus stacks and pairs running through the intercellular gaps (see Fig. R2 below), similar to what we show in our new Fig. 8.

Fig. R2 | *A. baumannii* AB5075-pGFP translucent and opaque colonies from the same Petri-dish both produce enough Csu pili to form pilus-linked clumps of bacteria. **a**, One of the two-days old translucent phenotype colonies picked for this experiment. Note the abrupt transition from dark brown central region to the translucent region close to the outer edge. **b**, One of the two-days old opaque phenotype colonies picked for this experiment. Here the change in colony darkness is far more gradual. **c–e**, Bacterial clumps (**c**) from a two days old translucent colony (**a**), which have lost most of their PNAG and eDNA during the suspension process (**d**), exposing the Csu pilus stack network in many places (**e**). **f–h**, Bacterial clumps (**f**) from a two days old opaque colony (**b**), which have lost most of their PNAG and eDNA during the suspension process (**g**), exposing the Csu pilus stack network in many places (**h**). Similar to the translucent colony, the opaque has multiple bacteria that produce an abundance of Csu pili, probably due to an early phenotype switch to the translucent phenotype in this part of the opaque phenotype colony.

We made the following change to “Growing *A. baumannii* and recombinant *E. coli* colonies for Csu pilus stack analysis” in the Methods section, page 17:

“We **initially** cultivated *A. baumannii* CCUG 19096T strain colonies using a 10-fold dilution series to achieve approximately 100–200 colonies per 0.5x LB agar plate (composed of 2.5 g NaCl, 5 g tryptone, 2.5 g yeast extract, and 8 g agar, pH 7.4, in 1 L of distilled water) to ensure the manifestation

of the translucent phenotype. We later observed that even seemingly opaque colonies had enough phenotype-switched cells to produce large pilus-linked clumps of bacteria.”

(5) L77-81: Authors show that stacks are not formed effectively in liquid cultures and are present in colonies. Would that not indicate that it is rather the accumulation of ECM (possible only under colony growth and not in liquids) that is primarily responsible for pili bundling and cell aggregation rather than the inherent pili-pili interaction which should be capable of forming even in liquid environments?

We thank the reviewer for this point. Csu pili do form stacks in liquid cultures leading to small clumps of bacteria with little to no ECM between them (Supplementary Figures 3 and 4a–d). However, these networks are far less extensive than in colonies, where surface adhesion, higher cell density, and possibly ECM facilitate stacking. To address this point as well as major comment 2.2 (see above), we performed experiments with PNAG-deficient (*pgaC::T26*) mutants and DNase I treatment (Fig. 8d–i) that show that bacterial clumps and rafts remain cohesive and interconnected by extensive Csu pilus stacks even in the complete absence of PNAG and eDNA. Together with prior deletion studies showing that biofilms fail to form without Csu pili despite intact ECM (see **Box 1** and our response to general comments), these findings indicate that pili–pili interactions are the primary driver of bacterial cohesion, while ECM plays only a supportive role.

We modified the text and added a new figure (Fig. 8). Please, see our response to major comment 2.2.

(6) L82-87: In the cryo-ET results (Fig. 3), there appears to be no continuity between pili from neighbouring cells to confirm that stacks are intercellular rather than intracellular, suggesting that the shown stacks originate from the same cell. Given the claims of exclusively anti-parallel stacking, how do the authors reconcile this observation?

In the cryo-ET experiment thickness of vitrified ice is critical for the quality of data (the thinner the better). Since regions with large and tight bacterial clumps had much higher thickness of vitrified ice, we focused on collecting data in a region between less tightly clumped bacteria. The main purpose of this cryo-ET experiment was not to depict a connection between two bacteria, but to reveal the morphology (flatness) of the stacks. The sample and grid preparation protocol for the cryo-ET experiment followed the exact same steps as when we prepared colony samples for negative stain TEM (with the only exception being the use of 1 % (w/v) Uranyl Acetate staining vs. cryo-plunging using Vitrobot IV). This means that the connections seen in the negative stain TEM images are the same as in the cryo-ET experiments. We have now added plenty of negative stain TEM images (see Fig. 2 and 8; Supplementary Fig. 1–4), showing both tight and more loose bacterial clumps connected by Csu pilus stacks as well as superstacks connecting multiple bacteria, and with these, we believe that this phenomenon should now be well documented.

We have now also highlighted that we used mostly the same sample preparation protocols for our negative stain TEM and cryo-ET samples by writing about them together in the following text from “Growing *A. baumannii* and recombinant *E. coli* colonies for Csu pilus stack analysis” in the Methods section, page 17:

“For the early negative stain transmission electron microscopy (TEM) experiments, the bacteria were grown for 16–20 hours at 37°C until the colonies reached a diameter of approximately 1.0–1.5 mm, and then stored at 8°C to keep them from growing further. Under this condition, colonies could be stored for 2–3 weeks, and remained usable even after three months with the only apparent change being the slow accumulation of amorphous materials, presumably PNAG or eDNA. For the later TEM experiments using *A. baumannii* AB5075 strain colonies and for cryo-ET using *A. baumannii* CCUG 19096T strain colonies, we grew them at 22°C for three days before using them immediately for the experiments without any storage.”

We similarly now highlight that the cryo-ET preparation method used suspended *A. baumannii* colonies in “Cryo-ET data collection and processing” in the Methods section, page 21:

“For cryo-ET sample preparation, *A. baumannii* colonies were suspended in a solution containing 10 nm colloidal gold beads (Sigma-Aldrich, Australia) pre-coated with 1% BSA.”

(7) L99-101: The authors state that a CsuE mutant with a compromised tip finger site impairs adhesion but does not affect Csu pili stacking, and consequently, biofilm formation remains unaffected. This statement requires quantification—biofilm formation levels should be explicitly compared to support the claim of unaltered levels.

We believe the reviewer’s question stems from our title for this section “*Csu pilus-mediated 3D biofilm assembly is independent of bacterial adhesion to surfaces*”. We admit that the title is confusing and invites a misunderstanding of us stating that CsuE has no role in the formation of 3D biofilms. We apologise for our poor wording. We have now changed the title to be more specific: “*Csu pilus-mediated bacterial interlinking and adhesion to surfaces depend on different structural elements of the pilus*”.

For *A. baumannii* 3D biofilms, both bacterial adhesion to surfaces and bacterial cross-linking are essential. In our previous study (Pakharukova et al., 2018), we demonstrated that mutations disrupting the hydrophobic tip finger site in the CsuE subunit impair adhesion and thereby prevent biofilm formation from the start. Within this framework of *A. baumannii* biofilm definition, the statement “*consequently, biofilm formation remains unaffected*” is not entirely accurate. In all classical biofilm assays that assess bacterial attachment to solid surfaces, the hydrophobic tip finger mutant yields negative results.

We believe the reviewer is, in fact, interested to know whether the CsuE hydrophobic tip finger site controls not only the bacterial adhesion to surfaces but also bacterial interlinking. Therefore, it is important to show that stack formation mediated by pilus rods can form and maintain a link between bacteria even without the CsuE contributing.

To address this, we investigated whether surface adhesion-deficient mutant Csu pili can cross-link recombinant *E. coli* into clumps. Negative stain TEM of liquid cultures from the mutant strain revealed Csu pilus-cross-linked clumps similar to those observed in native *A. baumannii* and wild-type recombinant *E. coli* strains, demonstrating that hydrophobic tip fingers are not required for bacterial cross-linking. This experiment further supports the conclusion that adhesion (monolayer formation) and 3D biofilm maturation (multilayer formation) are mechanistically distinct processes, mediated by two separate structural features of the Csu pilus—the hydrophobic tip fingers and the pilus rod, respectively.

In the revised version, we have added a new Supplementary Fig. 4 to demonstrate bacterial clump formation by the mutant in liquid cultures and in colonies. We also revised the text to describe this result explicitly and refined the terminology in the main text on page 5 to avoid potential confusion:

“This observation was further supported by the detection of similar Csu pilus stack-tethered bacterial clumps (or rafts) in liquid cultures of *E. coli* strain expressing recombinant Csu pili rendered unable to facilitate bacterial adhesion to hydrophobic surfaces due to a mutant tip-subunit CsuE with a compromised hydrophobic tip-finger site (Pakharukova et al, 2018) (Supplementary Fig. 4a–d). Similarly, extensive Csu pilus stacks formed in the induced colonies of the recombinant *E. coli* strain, regardless of whether it possessed the wild type or the mutant CsuE (Supplementary Fig. 4e, f). Collectively, these results indicate that Csu pilus-mediated surface adhesion and bacterial

interlinking are distinct processes that depend on different structural elements of the pilus. Nevertheless, both processes are essential for 3D biofilm formation.”

Additionally, wherever we refer to “biofilm” formed by Csu pilus rod stacking, we now specify that it is a “3D” biofilm, whenever it is necessary to ensure the clarity of the text.

(8) L112-115: The progressing nature of assembly has been only shown at low temperatures *in vitro* and therefore should not be mentioned confidently unless authors could show it through *in vivo* imaging. Moreover, the assertion that the biofilm-like structures observed *in vitro* indicate that biofilm formation is a “spontaneous process inherently programmed within the structure of the pili” is strong but lacks consideration of alternative explanations. Could environmental factors, such as salinity or pH, also contribute to stack formation? Testing whether stacks still form under high ionic strength conditions would help exclude non-specific interactions driven by low ionic strength. Additionally, is stacking reversible under certain environmental conditions?

We thank the reviewer for this excellent suggestion. To address this point, we thoroughly investigated Csu pilus stack formation *in vivo* at different temperatures, salinities, and pH levels.

The effect of temperature has already been discussed above. We analysed stack formation in *A. baumannii* colonies immediately after growth at 22°C, 30°C, and 37°C using negative stain TEM (new Supplementary Fig. 2b–c). At all three temperatures, we observed identical and extensive networks of pilus stacks using the hypervirulent strain AB5075 of *A. baumannii*. These results demonstrate that *in vivo*, Csu pilus stacking networks form very rapidly and in parallel with colony growth, and do so robustly across a wide range of both ambient and physiological (body) temperatures.

We also tested the effect of pH (ranging from 6.0 to 8.5) and salinity (from 42.8 mM to 200 mM NaCl), and observed no noticeable effect on Csu pilus stack formation or bacterial clumping *in vivo* (new Supplementary Fig. 2d–g). However, growth at a high salt concentration of 400 mM significantly affected *A. baumannii*. Under this condition, the colony diameters were only a fourth of the ones grown at 42.8 mM NaCl, the bacteria expressed fewer Csu pili, and the resulting bacterial clumps were smaller and less dense. Nonetheless, Csu pilus stacks still formed wherever there were sufficient numbers of Csu pili near one another, highlighting the strong intrinsic tendency of these pili to assemble into stacks even under suboptimal conditions.

We have inserted the following sentence into the main text, page 4:

“Stacks may contain more than 15 pili, with pili often forming lateral interactions extending over half of their length (Supplementary Fig. 1a–c). Superstacks, continually incorporating new pili from the surfaces of multiple bacteria, can exceed 12 μm in length (Supplementary Fig. 1d–h), clumping bacteria in 3D structures (Supplementary Fig. 2). These structures formed equally efficiently under both ambient and physiological temperatures (22–37°C) and across a broad range of pH (6.0–8.5) and salt concentration (42.8–400 mM) (Supplementary Fig. 2).”

Since Csu pilus stacks form spontaneously *in vivo* in such short amounts of time (less than 3.5 hours after induction; see Supplementary Fig. 4) that observing their actual progress would be difficult, we also adjusted the sentence on page 5:

“The striking similarity between the *in vitro* and *in vivo* networks, along with the progressing nature of their assembly *observed in vitro*, suggests that 3D biofilm formation is a spontaneous process inherently programmed within the structure of the pili.”

(9) L135-141: During grid preparation and freezing, did the authors observe any dependence of stacking on ice thickness? Could the thin ice required for SPA induce or perhaps also limit the formation of Csu pili stacks?

No, we did not observe any dependence of stacking on ice thickness. Stack formation appears to be a highly robust process, as it was unaffected by any of the conditions we tested (see our answer above). Moreover, the stacks used for single-particle cryo-EM had already formed well before being added to the cryo-EM grid, as they were visible as a gel at the bottom of the sample tube (Extended Data Fig. 4a). We also confirmed that this gel consisted of extensive Csu pilus stacks using negative stain TEM (Fig. 4d). Obtaining the same result with two different methods rules out any possible influence of vitrified ice thickness on stack formation.

(10) L189-202: I strongly encourage the authors to provide mutational validation for their observations regarding CupE pili. This would significantly strengthen the novelty of their claims, allowing for extrapolation of their findings to all archaic pili based on experimental data, increasing impact.

As discussed in point 1.2, the technical challenges associated with expressing CupE pili place experimental work on CupE pili beyond the scope of the present manuscript; this will be addressed in future studies.

We have added the following text in Discussion (pages 11–12):

“Our results support the dominant role of archaic Csu pili in clumping bacteria into 3D structures. However, a role for other EPS, such as PNAG and eDNA, at different stages of this process cannot be excluded. Similarly, several protein surface receptors, including OmpA (Gaddy *et al.*, 2009), Ata (Bentancor *et al.*, 2012) as well as Biofilm-Associated Protein (Bap) and its closely related Blp1 and Blp2 (Brossard & Campagnari, 2012; De Gregorio *et al.*, 2015) facilitate biofilm formation. As these receptors have smaller structures, they likely mediate more localized interactions compared to the Csu pili. In addition, *A. baumannii* produces Type IV pili, and its different variants promote motility in AB5075 strain and biofilm formation in ACICU strain (Ronish *et al.*, 2019). Interestingly, the latter strain has a transposase gene inserted between *csuA/B* and *csuA* within its *csu* operon. In our colonies of *A. baumannii* 19096T and AB5075, we observed little to no filaments wide enough (72 Å) (Wilharm *et al.*, 2013) to be interpreted as Type IV pili. Only colonies grown at 30 °C and 37 °C displayed a few bacteria with such filaments, while most cells expressed abundant Csu pili. This agrees with the observation that cyclic AMP acts as a master regulator, suppressing the expression of Csu pilus subunits, PNAG-synthesizing enzymes and quorum sensing, and thus biofilm formation in general, while strongly upregulating Type IV pilus subunits and other motility-related genes (Harkova *et al.*, 2024). Therefore, while Type IV pili may contribute to biofilm formation by mediating adhesion to certain surfaces, they likely lose relevance once *A. baumannii* switches to producing Csu pili. However, the expression regulation could be different in other species. Studies of other archaic CUP pili, such as the *P. aeruginosa* CupE system, may help clarify this.”

(11) L204: The claim that PNAG embeds pilus stacks in mature biofilms is based on WGA labelling. To confirm this, could the authors treat samples with PNAG-hydrolyzing enzymes and analyze the effect via TEM imaging? Additionally, how do biofilm levels compare in the presence and absence of this enzyme? Could authors discuss the possibility that trapping PNAG and eDNA is the main involvement of Csu pili in biofilm formation rather than a side effect of stacking.

This is an excellent suggestion. Please see our extended answer to a related point 2.2. We attempted to purchase the PNAG-hydrolysing enzyme Dispersin B from Kane Biotech Inc., but they

refused to sell it to us due to regulatory restrictions. Fortunately, we were able to implement the second approach suggested by the reviewer earlier, mutagenesis. Specifically, we obtained an *A. baumannii pgaC::T26* mutant, which is unable to produce PNAG because the glycosyltransferase gene *pgaC* is disrupted by a T26 transposon. We then grew this mutant and suspended three-day-old colonies in DNase I solution to degrade eDNA. After dilution and washing, the remaining bacterial clumps, or “tight rafts” (Fig. 8g), were highly organized and resembled wild type clumps, but displayed clear gaps devoid of any material resembling PNAG or eDNA (Fig. 8h), while extensive Csu pilus stacks were observed traversing these gaps (Fig. 8i). The fact that the clumps retained their macroscopic structure in the absence of PNAG and eDNA demonstrates that these components are not essential for maintaining bacterial cohesion.

We have inserted the following sentences into the main text on page 9:

“While most bacteria seen in our negative stain TEM samples formed dense clumps with aplenty of Csu pilus stacks on the surfaces and PNAG or eDNA filling the tight gaps (Fig. 8a–c, Supplementary Fig. 2), others formed loose, yet cohesive pilus-linked rafts with otherwise empty, wide gaps (Fig. 8d–f). The distinctive lack of PNAG and eDNA in these rafts indicates that these EPS contribute little to clump cohesion. To test this further, we grew *A. baumannii* AB5075 mutant *pgaC::T26*, which is unable to produce PNAG (Gallagher *et al.*, 2015), and treated suspended colonies with DNase I and the DNA-intercalating agent chloroquine to degrade eDNA (Buzzo *et al.*, 2021). Subsequent dilution and washing steps revealed several highly organized rafts, with tight, clear gaps and extensive Csu pilus stacks traversing them (Fig. 8g–i), further supporting the conclusion that Csu pilus stacks link the bacteria in early 3D biofilms independently of PNAG and eDNA.”

(12) L237-240: Are there any documented differences in the production of Csu pili in liquid vs biofilm growth that could support this?

We assume the reviewer is referring to the following statement: “Second, the small gaps between neighbouring cells in a biofilm help position Csu pili antiparallel to each other, promoting stack formation (Fig. 7). Csu pilus stacks between daughter cells and neighbouring cells could help maintain their spatial orientation, allowing the biofilm to grow taller without breaking apart. In liquid cultures, such neighbouring cells are most often absent (Supplementary Fig. 8).”

We refer to such study (ref. 39, Rumbo-Feal *et al.*, 2013) and discuss this issue a few sentences above:

“Although this process can occur both in liquid and solid media, the formation of stacks linking bacteria together is dramatically more efficient in 3D biofilms than in liquid cultures. *A. baumannii* in biofilms produce more pili than liquid culture cells (Rumbo-Feal *et al.*, 2013), but since clumps are also rare in *E. coli* liquid cultures overexpressing Csu pili, the higher number of pili is unlikely the main factor here.”

Our conclusion is largely based on observations from negative stain TEM analysis of suspended colonies, where it is widespread that extensive Csu pilus stacking networks follow the curvature of bacteria some distance away from the cell surface (Fig. 2 and Supplementary Fig. 1d–h, see also Fig. R2e, h above). These superstacks are consistently present in samples from both *A. baumannii* and *E. coli* colonies and are most visible on the exposed surfaces of the bacteria. In contrast, in our liquid culture samples, both from *A. baumannii* and *E. coli*, the Csu pili tend to point in every direction with only a few stacking interactions. However, in the bacterial clumps found in liquid cultures, while the Csu pili on the outer surfaces still point directly outwards, the Csu pili between the bacteria show stacking interactions (Supplementary Fig. 3 and 4) with many of them running parallel to the cell surfaces, especially when the gaps are narrow (Supplementary Fig. 3).

(13) L251-252: Has it been tested—either in this study or elsewhere—whether non-zigzagged pili are unable to form gel-like structures after prolonged incubation at low temperatures? If not, this statement should be softened.

The reviewer is referring to the following statement: “We show that such flat assemblies require helical filaments with very short repeats, suggesting that the zigzag architecture of archaic CUP pili has likely evolved to mediate 3D biofilm formation.”

Csu pilus stacks do not require “prolonged incubation at low temperatures” to form *in vivo*. On the contrary, stack formation is a surprisingly rapid process that occurs simultaneously with 3D biofilm growth across a wide range of temperatures (see our answers above). We propose that the zigzag architecture provides multiple features beneficial for 3D biofilm formation, including the unusually short helical repeat that enables a high density of binding junctions between pili. Classical and alternative CUP pili, by contrast, have different architectures and are typically not involved in 3D biofilm formation; instead, they generally mediate highly specific adhesion to biological targets. Nevertheless, one example of classical CUP pili contributing to 3D biofilm formation has been thoroughly documented (*E. coli* common pili), and it is quite possible that other classical or alternative CUP systems have also been adapted for bacterial cross-linking in 3D biofilms. Documented cases in which both CUP and non-CUP pili participate in 3D biofilm formation are discussed in the Discussion section on page 12:

“Archaic zigzag pili represent the first example of using highly structured, regular flat sheets for 3D biofilm assembly. Several other EPS scaffolds reported in bacteria and archaea are also DSC-linked fibres, but they mostly form twisted (Garnett *et al.*, 2012),(Gohl *et al.*, 2006) or cable-like bundles (Bohning *et al.*, 2022; Gaines *et al.*, 2022; Sonani *et al.*, 2025; Wang *et al.*, 2022). *E. coli* common pili, assembled through classical CUP mechanisms (Garnett *et al.*, 2012), and archaeal bundling pili (ABP) (Wang *et al.*, 2022) from *Pyrobaculum calidifontis* involve antiparallel interactions, while archaeal Tafi pili from *Natrinema* sp. J7-2 appear to involve both parallel and antiparallel interactions (Sonani *et al.*, 2025). The directionality remains unknown for *Acinetobacter* sp. strain BD413 classical Acu pili (Gohl *et al.*, 2006), *Sulfolobus acidocaldarius* archaeal thread filaments (Gaines *et al.*, 2022) and TasA pili of the Gram-positive bacterium *Bacillus subtilis* (Bohning *et al.*, 2022). Further studies are necessary to understand how these bundles spontaneously assemble to support 3D biofilms.”

(14) L269-270: Structurally, if one site on the pilus rod is occupied by another to form a stack, why would the remaining binding sites become inaccessible for another junction? Can the authors confidently exclude the possibility of three-dimensional stacking instead of exclusively flat stacking? Is there any *in vivo* evidence supporting the formation of flat connections rather than thick, circular bundles? If flat stacks were the predominant arrangement, wouldn't they be visible in SEM images? While the cryo-ET data suggests flat stacks, without subtomogram averaging, this conclusion remains uncertain.

We split the comment in two parts and address each part separately below.

(14.1) Structurally, if one site on the pilus rod is occupied by another to form a stack, why would the remaining binding sites become inaccessible for another junction? Can the authors confidently exclude the possibility of three-dimensional stacking instead of exclusively flat stacking?

We thank the reviewer for raising this important point. To address it, we created a new Supplementary Fig. 7, which we believe will help readers understand both the geometrical constraints and the range of possible structures.

Theoretically, despite the geometric constraints outlined in the new Supplementary Fig. 7a, a wide variety of structures are possible (Supplementary Fig. 7b), including three-dimensional assemblies (Supplementary Fig. 7e, f). However, despite extensive analyses using negative stain TEM, single-particle cryo-EM, and cryo-ET, and examining Csu pilus stacks in both *in vivo* (colonies, liquid cultures) and *in vitro* (extracted pili) contexts, we have never observed any structures that could be confidently interpreted as 3D stacks. This stands in sharp contrast to the abundant evidence for pairs and flat stacks under both conditions. Thus, while 3D stacks remain a theoretical possibility, they are likely to be extremely rare.

A plausible explanation for their rarity is that such structures would compromise the remarkable stretching capacity of individual Csu pili (Pakharukova et al., 2022). As shown in the current Supplementary Fig. 13a–d, the ability of a single Csu pilus to nearly double its length by opening its clinch-structures is largely preserved in pilus pairs and in both types of flat stacks, with type 2 stacks potentially also allowing lateral stretching. In contrast, none of the predicted 3D stacks would accommodate this property. For example, the helical tube variant of the type 1 stack lacks subunits that can stretch without disrupting multiple binding junctions (Supplementary Fig. 7e; end view), making it not only extremely rigid but also inhospitable to slightly pre-stretched pili attempting to join the structure. Similarly, the second type of 3D stack contains only two stretchable subunits per seven-subunit repeat (Supplementary Fig. 7f; orange and green subunits). While this configuration would allow limited extension, it would still be far more rigid than flat stacks or pairs.

It is also important to consider the abundant “random bundles” frequently observed in our samples. These are twisted, disorganized versions of flat stacks and pairs, sometimes stabilized by additional pili wrapping around them and forming junctions that help maintain their orientation (Fig. 4b). Occasionally, a pilus may transiently contact three antiparallel pili simultaneously—similar to the geometry of predicted 3D stacks (Supplementary Fig. 7f)—but these interactions never give rise to regular 3D architectures. Instead, we consistently find only pairs and flat stacks embedded within such bundled regions. Competition with these disorganized random bundles is therefore another likely reason why highly ordered 3D stacks are so rare.

We have added the following sentences in Discussion (page 11):

“This extensive stretching capability may explain why Csu pili tend to form only flat stacks, even though the available binding sites theoretically allow for much greater variation (Supplementary Fig. 7a, b). It appears that, apart from the wavy type 1 stack variant that may have a somewhat reasonable ability to stretch (Supplementary Fig. 7c), the spacing of binding junctions in others would exclude most of the subunits from stretching (Supplementary Fig. 7d–f). Importantly, the high elasticity of stacks would make established 3D bacterial clumps more resistant to turbulent environments by absorbing stress. Interestingly, the stacks embedded into extracellular material made of PNAG and eDNA are reminiscent of reinforced concrete, as in this combination, the stack may act like steel bars or mesh, providing resistance to tensile and shear stresses, while PNAG or eDNA would provide resistance to compression. Further experimental biophysical studies will be required to investigate the predicted properties.”

(14.2) Is there any *in vivo* evidence supporting the formation of flat connections rather than thick, circular bundles? If flat stacks were the predominant arrangement, wouldn't they be visible in SEM images? While the cryo-ET data suggests flat stacks, without subtomogram averaging, this conclusion remains uncertain.

As stated earlier, our cryo-ET and negative stain TEM images of bacterial colonies and liquid cultures were obtained from live bacteria grown at 22–37 °C, with sample preparation steps kept to a minimum to reduce the risk of artefacts. These images therefore depict the bacteria in their native *in vivo* state. Both techniques provide clear evidence of flat stacks: cryo-ET as 3D structures and negative stain TEM as rows of regularly spaced pili that fray into individual pili at their ends.

By contrast, the limited SEM images we included are not representative of stack structures. Their purpose was primarily to provide an overview and to visualize PNAG accumulation. For more comprehensive SEM data on Csu pili in *A. baumannii* biofilms, earlier studies by Tomaras et al. (2003) and Ahmad et al. (2019) are more informative, showing that stacks can be far more extensive than in our selected image. Moreover, SEM is a low-resolution technique requiring extensive sample preparation, which is well known to introduce artefacts. Since Csu pilus stacks form in a hydrated environment, drying and coating samples with heavy metals is highly likely to distort their native architecture (e.g., flat stacks could roll into tube-like structures or collapse under the stain). Additionally, SEM provides only 2D projections and therefore cannot reliably resolve 3D structures. In contrast, cryo-ET produces complete, rotatable 3D segmentations of structures preserved in vitrified liquid, capturing them in their native state. Our cryo-ET data were obtained from freshly grown colonies, suspended and applied to grids within minutes of removal from Petri dishes, leaving little to no opportunity for artefacts to arise. For these reasons, cryo-ET provides far more reliable evidence of stack morphology than SEM.

Regarding the reviewer's suggestion that subtomogram averaging is required to validate our cryo-ET data, we would like to point out that, for subtomogram averaging to be meaningful, it is necessary to align and average large numbers of near-identical particles (or helical assemblies) to obtain higher resolution (often nanometre or sub-nanometre) structures. Csu pilus stacks are too sparse to provide a sufficient number of near-identical subvolumes for subtomogram averaging, rendering this method unsuitable. Nevertheless, in all our cryo-ET reconstructions, the stacks are consistently several-fold wider than they are thick, as demonstrated in Supplementary Video 1. Moreover, we observed these flat stacks at various orientations relative to the imaging plane, with individual pili visible in the same tomograms, making it implausible that the observed stacks are imaging artefacts. To further support this conclusion, we have added a series of tilt images of our segmentation (Fig. 3i), showing a single stack rotated 90° in four steps, directly illustrating its flatness.

(15) L233-236: The authors suggest that increased turbulence explains why liquid cultures do not exhibit the same levels of Csu-mediated biofilm formation. Could they provide evidence demonstrating the impact of growth conditions on Csu stack formation *in vitro*? Additionally, could they visualize the effect of turbulence on pre-assembled Csu stacks?

Again, an interesting point and excellent direction for future studies. We have partially addressed this point with our new *in vivo* experiments (1) and some *in vitro* observations (2):

(1) The effect of environmental conditions on Csu pilus assembly is much easier to test *in vivo* as forming them *in vitro* takes significantly more time, possibly due to turbulence or lack of narrow bacterial gaps guiding the pilus orientations. As we mentioned in our answer to the reviewer's comment 8, we conducted an experiment of growing *A. baumannii* colonies on standard (0.5x LB, 42.8 mM NaCl and pH 7.4, 3 days at 22°C) and adjusted conditions, changing NaCl concentrations (200 mM and 400 mM) and pH values (pH 6.0 and 8.5) (Supplementary Fig. 2a, d–g). We also grew colonies at 30°C and 37°C overnight to confirm the effect of temperature and time on Csu pilus stack formation (Supplementary Fig. 2a–c). Our results indicated that Csu pilus stacks form rapidly in all tested conditions with the only limiting factor being high NaCl concentrations that reduce bacterial growth and the number of Csu pili per bacteria. Even then, stacks formed when enough of pili were sufficiently close to one another, so the excess NaCl did not affect the stack formation itself.

(2) When it comes to the effects of turbulence on pre-assembled Csu pilus stacks *in vitro*, the process of pipetting stacks that accumulated at the bottom of a fraction tube (Supplementary Fig. 5a) and adding them onto a TEM grid is typically enough to temporarily resuspend most of the visible accumulated substance into the solution. However, it seems insufficient to break the stacks themselves, as the negative stain TEM grids made from such samples tend to have aplenty of regular stacks (Fig. 4c, d). The resuspended stacks will also settle back to the bottom of the tube, allowing us to use the same stack sample as long as enough stacks remain, which was the case, when we checked for the presence of stacks using negative stain TEM (Fig. 4d) before using the sample to make our cryo-EM grids for structure determination (Supplementary Fig. 5). Therefore, the Csu pilus stacks are quite stable, once they have formed.

We believe that analysing the role of turbulence in stack formation requires a dedicated study using appropriate methods. Such work could potentially suggest new strategies to prevent or disrupt the formation of Csu pilus-mediated 3D biofilms and may have considerable medical relevance.

Reviewers 2 and 3

We are very pleased that the reviewers found our study to be “rigorous, well-written, well-presented, and of interest to a broad scientific readership,” and that they supported our interpretation of most cryo-EM structures. We are grateful for the reviewer’s thorough assessment and for the many excellent suggestions and corrections, which have helped us to improve the manuscript. In response, we have improved the description of our structure refinement protocol, added the missing FSC curves, examined the fit of individual subunits into the pilus pair map, performed 3D variability analysis, and prepared a detailed workflow figure. To further enhance clarity and interpretation of our negative stain TEM and cryo-ET data, we improved several figures, added four new figures illustrating different aspects of Csu pilus stack networks, revised the figure legends, and included additional references. Finally, we re-examined both old and new samples to assess the presence of Type IV pili, and our data indicate that Csu pili and Type IV pili are not co-expressed, as discussed in the revised manuscript.

Reviewer #2 (Remarks to the Author):

Reviewer #3 (Remarks to the Author):

I. Overall review:

The archaic chaperone usher Csu pili formed by *Acinetobacter baumannii* are known to play essential roles in the formation of biofilms. Csu pili have been previously shown to biofilms on abiotic surfaces as well as biofilm pellicles and play crucial roles in adhesion to mammalian tissues. This study by Malmi et. al uses a variety of electron microscopy techniques (SEM, Negative stain TEM, Cryo-ET, Cryo-EM, and gold conjugation labeling) to investigate the role that Csu pili play in biofilm formation at multiple length scales from the micro/nanoscale of SEM and cryo-ET to sub-nanometer and atomistic details provided by single particle cryo-EM. Using these length scales the authors are able to strongly suggest how the Csu pili are involved in biofilm formation. The cryo-EM structural details of the pilus pairs and stacks, while low-resolution, are able to show novel anti-parallel packing interactions between pili. When attached to cells these anti-parallel interactions would presumably mostly occur between different cells which is plausible. Upon examination of the atomic models and cryo-EM density maps, I am confident in the authors’ interpretation of most of their cryo-EM structures. Any concerns are noted in the next section of my review. With all of this data the authors propose a plausible model for Csu pili-mediated biofilm formation.

Overall, this study is rigorous, well-written, well-presented, and should have appeal to a broad scientific readership. Upon sufficient response to my concerns from the authors, I think this paper would be an excellent fit for publication at Nature Communications. I have several points which I would strongly like to see the authors address as well as some minor suggestions that could help improve their story/results.

II. Major points to be addressed:

1. In the Methods on page 16 lines 30-31 it is stated: “Homogeneous Refinement using our earlier 8.5 Å map (unpublished) of type 1 stack revealed that there are two types of stacks”. This seems very unlikely that two types of stacks were revealed this way. Do the authors mean “Heterogenous Refinement”?

We thank the reviewer for this observation. We first observed indications of two distinct stack types during Homogeneous Refinement (see Fig. R3 below), which were subsequently confirmed through

multiple rounds of Heterogeneous Refinement (now outlined in the new workflow figure, Supplementary Fig. 17).

The Csu pilus pairs are nearly identical in both stack types, differing only in how they bind to additional pili (Fig. 5b, Supplementary Fig. 7a, b). As a result, refinement programs often overlapped one pair from each type, producing a combined density map where the two stacks intersected at 51.43° ($360^\circ/7$) angle. In this overlap, type 1 stacks could be recognized by 154.43° ($360^\circ/7 \times 3$) angles between three adjacent pili, whereas type 2 stacks had 102.86° ($360^\circ/7 \times 2$) angles. The sharper geometry of type 2 stacks was evident from the end views, and type 1 stack models did not fit well into type 2 stack densities. Importantly, the combined maps showed unequal density distribution, consistent with two distinct flat stacks being superimposed rather than the pili forming a single, unified 3D stack (Supplementary Fig. 17e and 7f; Fig. R3b). By fitting modified type 1 stack overlap models into these combined maps, we confirmed that type 2 stacks use alternative subunits for pilus–pilus contacts rather than using different types of binding junctions altogether (Fig. R3c, d). This density pattern was not unique to this initial refinement run but recurred whenever we used unsorted particle sets, and the final map orientations (Supplementary Fig. 17h) still reflect this characteristic $\sim 51.43^\circ$ overlap.

Thus, although the job in question was a Homogeneous Refinement, the presence of two nearly identical pilus pairs caused the algorithm to merge features from both stack types, revealing their coexistence in the dataset. This overlap, together with pseudosymmetry from 180° flipped orientations, represent the two main sources of heterogeneity in our stack density maps (see response to point 4).

Fig. R3 | Detecting two stack patterns from a Homogeneous Refinement map produced using unsorted particles, a 8.5 Å resolution type 1 stack search model and no mask. **a**, An initial 8.5 Å resolution map obtained from an older dataset (Supplementary Fig. 17a) shows the 154.43° angles between three adjacent pili and the neighbouring binding junctions overlapping by a single subunit, which are typical of a type 1 stack structure. We used this map as a search model for a Homogeneous Refinement job (**b–d**) using unsorted particles from the new dataset (Supplementary Fig. 17b) and no mask. **b**, End view from a Homogeneous

Refinement map produced using unsorted particles and a 8.5 Å resolution type 1 stack search model (Supplementary Fig. 17e). The vertical row of pili marked with a blue rectangle corresponds to a type 1 stack density, whereas the diagonal row of pili marked with a red rectangle corresponds to the previously unknown type 2 stack. Csu pilus type 1 and 2 stacks, extended by overlapping copies of the published structures, fit well to their corresponding map densities. The central pair shared by the two types of stacks is within the region, where the two rectangles cross each other. **c**, Side view of the type 1 stack corresponding to the blue rectangle in **(b)** with its right side marked next to the rectangle in **(b)**. Csu pilus type 1 stack model fits to the corresponding density pattern reasonably well, though particles flipped 180° due to stack pseudosymmetry weaken the binding junction densities. **d**, Side view of the type 2 stack corresponding to the red rectangle in **(b)** with its right side marked next to the rectangle in **(b)**. Csu pilus type 2 stack model fits to the corresponding map better than type 1 stack, as type 2 stack has fewer issues with pseudosymmetry.

2. For the Csu pili pairs, the results section on page 5 lines 30-32 state that: “Since the pilus pair lacks helical symmetry, we generated an initial model based on the structure of the single pilus rod by optimizing the overlap between its projections and a 2D class average (Supplementary Fig. 10–12)”. This model was built prior to reconstruction of the pilus pairs map and then docked into the map. Do the authors find any significant differences if they build the model by docking individual subunits into the low-resolution map?

We thank the reviewer for the suggestion. We docked individual donor-strand complemented CsuA/B subunits into the Csu pilus pair density map. The overall result was very similar to docking the full pilus pair model (Fig. R4a, b).

Fig. R4 | Difference between fitting the full model versus individual subunits to Csu pilus pair map. a, The published Csu pilus pair model fit inside the Csu pilus pair map. The angle between the pili in model is based on Csu pilus pair 2D class overlap model (Supplementary Fig. 14c, d), which also fits to the Csu pilus stack maps without issues. **b,** Individual CsuA/B subunits from the model without their own donor strand (cutting the peptide bond between Thr14 and Gly15) and complemented by the donor strand from the following subunit (CsuA/B-dsA/B) fit inside Csu pilus pair map. The subunit orientations are mostly similar to the published pair model, though the lower density at the pilus periphery and map noise has caused the individual subunits to pull back from the pilus edges towards the central axes of the two pili. The individual subunits ignoring their chemical constraints has also resulted in clashes between the subunits (not shown) as well as distances too long to reconnect Thr14 and Gly15 at the cut donor strand linker. **c,** The best 2D classes for the Csu pilus pair structure based on manually picked particles. The last 2D class is the one used for overlap analysis (Supplementary Fig. 14–16). While these 2D classes show intricate details, the lack of alternative views made it difficult to construct a 3D map and ultimately left it noisy with low resolution.

3. Map:Map FSC curves for each of the structures from cryoSPARC should be presented as supplementary figures with resolution and the either the 0.143 or 0.5 FSC line being indicated.

We thank the reviewers for the suggestion. We have now added the FSC curves for each structure (Supplementary Fig. 6).

4. Workflows for each cryo-EM structure should be presented so readers can easily follow how the maps were generated. These should start with particle picking and show each step in the reconstruction process (heterogeneous refinement, homogeneous refinement, post processing, .etc).

We thank the reviewers for the suggestion. We have made a new figure (Supplementary Fig. 17) showing the map refinement workflows for the Csu pilus rod segment, the Csu pilus rod pair, the initial map of Csu pilus type 1 stack as well as the combined map refinement workflow for Csu pilus type 1 and 2 stacks. The first three workflows (Supplementary Fig. 17a) are quite standard, but the workflow for pilus stacks 1 and 2 is more complex (Supplementary Fig. 17b–h), as explained below:

The reconstruction of pilus stacks type 1 and type 2 faced two main issues. First, Csu pilus pairs were nearly identical between the two stack types, and since each pair constituted a large portion of the map, refinement programs frequently misassigned particles. This produced composite maps in which density patterns from the two stacks crossed at 51.43°, while non-shared junctions showed weaker density. Second, particle sets contained mixtures of 180° flipped orientations, as the pillar-like stacks and zigzag features of individual pilus rods appeared similar when inverted. This pseudosymmetry caused overlaps during refinement, evident in 2D classes showing adjacent sharp and diffuse regions (Supplementary Fig. 17d).

To separate particles from the two stack types, we ran iterative Heterogeneous Refinement job pairs, pooling particles assigned to the same type for subsequent rounds. Residual misassignments were addressed by subdividing type 1 stack particles into seven subsets and excluding the two poorest based on Homogeneous Refinement results. For type 2 stacks, we instead used a type 1 model as a decoy class to clean the set, as type 1 stack maps appeared less prone to snatch type 2 stack particles than *vice versa*.

Pseudosymmetry was further resolved by using multiple 180°-flipped versions of the stack maps as search models in Heterogeneous Refinement, while deliberately lowering the O-EM learning rate so occasional flipped particles would not bias the maps (Supplementary Fig. 17g). After refinement, we rotated results to match the original model and combined them in Local Refinement to avoid orientation randomization (Supplementary Fig. 17h).

5. The model in Figure 7 proposed for Csu pili-mediated biofilm formation focuses on cell division as the means from which more cells are introduced into this biofilm. *Acinetobacter baumannii* have been shown to have T4P mediated twitching motility. Could this also significantly contribute to biofilm formation? If so, such considerations should be introduced into the paper. If not, I would suggest that the authors also address this briefly in the manuscript discussion.

We thank the reviewers for raising this point. To address it, we re-examined all our old and new negative stain TEM micrographs of *A. baumannii* colonies from two strains (CCUG 19096 and AB5075) grown under different conditions (temperature, pH, salt concentration, and incubation time). We tested colonies of the translucent phenotype (most samples), mixed phenotype (AB5075 during revision), and opaque phenotype (see Fig. R2 in our responses to Reviewer 1). Type IV pili in *A. baumannii* (~72 Å in diameter; (Wilharm *et al.*, 2013)) are notably wider than Csu pili, which vary from ~55 Å at their widest point to less than half of that (Fig. 4b), making their average width around half the width of Type IV pili. In most TEM samples, we observed no filaments wide enough to be identified as Type IV pili. Only in colonies grown at 30°C or 37°C did we detect a few filaments too wide to be Csu pili (Fig. R5a), which could possibly represent Type IV pili. By contrast, all colonies displayed abundant Csu pili forming extensive stacks. These findings suggest that in colonies, Csu pili overwhelmingly dominate, and the rare appearance of Type IV pili likely reflects a few cells switching to (or retaining) the opaque phenotype.

This result is consistent with prior studies showing that Csu and Type IV pili are antagonistically regulated. Recently, (Harkova *et al.*, 2024) demonstrated that cyclic AMP (cAMP) acts as a master regulator controlling the switch between two phenotypes, affecting ~234 differentially expressed genes (see their Fig. 2–3). Specifically, cAMP suppresses expression of Csu pilus subunits, PNAG biosynthetic enzymes and quorum-sensing, and thus biofilm formation overall, while strongly upregulating Type IV pilus subunits and other motility genes. Thus, Csu pili and Type IV pili are not expected to be co-expressed: Csu pili promote sessile biofilm growth, while Type IV pili mediate twitching motility—processes that are physiologically incompatible. This conclusion is also supported by (Giles *et al.*, 2015), who showed that cAMP degradation by phosphodiesterase CpdA increased CsuA/B expression and promoted surface pellicle formation, and by (Nait Chabane *et al.*, 2014), who identified CsuA/B as the most abundant protein in the pellicles. The cAMP-dependent switch likely underlies the opaque–translucent phase variation, with the translucent phenotype producing more Csu pili and displaying reduced motility (Ahmad *et al.*, 2019; Tipton *et al.*, 2015).

Although Type IV pili appear dispensable in colonies, they may contribute more significantly to pellicle formation in liquid cultures, which may better reflect conditions for motile, opaque-phase bacteria. However, all our liquid culture experiments used translucent-phase stocks to ensure efficient Csu pilus expression, and we currently lack opaque-phase stocks. Furthermore, it would be difficult to determine whether Type IV pili in pellicles derive from pre-existing opaque cells or from phenotype switching during growth. Likewise, small clumps of cells with Type IV pili in liquid culture could reflect phenotype switching rather than true accumulation. It is also possible that motile cells lose their Type IV pili upon joining a pellicle. While we recognize this as an interesting direction for future work, it is beyond the scope of the present study.

As an informative comparison, our TEM samples of *Pseudomonas aeruginosa* PA14 colonies grown under similar conditions at 22°C showed bacteria with multiple flagella and abundant wide filaments consistent with Type IV pili (Fig. R5b), with few other pili present. This suggests that under our standard growth conditions, the equilibrium in *P. aeruginosa* strongly favours the motile phenotype.

Fig. R5 | Type IV pili in *Acinetobacter baumannii* and *Pseudomonas aeruginosa* colonies and their absence among pili purified from *E. coli* pBAD-Csu overexpression system. **a**, Possible Type IV pili observed on *A. baumannii* AB5075 colony grown at 30°C overnight. There were no filaments wide enough to interpret as Type IV pili in any of our liquid cultures or colonies grown at 22°C. While colonies grown at 30°C and 37°C did have a few bacteria that seemed to express Type IV pili, these were rare, and the vast majority of bacteria produced Csu pili assembling into extensive stacks (Supplementary Fig. 2b, c). **b**, *P. aeruginosa* PA14 grown on identical 0.5x LB agar plates as *A. baumannii* formed rapidly spreading, flat colonies with the bacteria covered in flagella as well as numerous filaments that we believe are Type IV pili due to their width, while producing very few other types of filaments. This suggests that under these growth conditions, *P. aeruginosa* strongly exhibits a motile phenotype. Since CupE are necessary for the formation of stationary, mushroom-like 3D biofilms in *P. aeruginosa* (Giraud *et al.*, 2011), their expression is likely suppressed in these motile phenotype cells. **c**, Csu pilus fractions purified for cryo-EM from *E. coli* pBAD overexpression system using a Source30Q anion exchange column and then checked with SDS-PAGE for purity. Since the Csu pili were produced in *E. coli* and then purified with an ion exchange column, there were very few other types of proteins present in the fractions used for cryo-EM. Naturally, none of the cryo-EM processing steps showed any Type IV pili.

We have added the following text to the Discussion section on pages 11–12:

“Our results support the dominant role of archaic Csu pili in clumping bacteria into 3D structures. However, a role for other EPS, such as PNAG and eDNA, at different stages of this process cannot be excluded. Similarly, several protein surface receptors, including OmpA (Gaddy *et al.*, 2009), Ata (Bentancor *et al.*, 2012) as well as Biofilm-Associated Protein (Bap) and its closely related Blp1 and Blp2 (Brossard & Campagnari, 2012; De Gregorio *et al.*, 2015) facilitate biofilm formation. As these receptors have smaller structures, they likely mediate more localized interactions compared to the Csu pili. In addition, *A. baumannii* produces Type IV pili, and its different variants promote motility

in AB5075 strain and biofilm formation in ACICU strain (Ronish *et al.*, 2019). Interestingly, the latter strain has a transposase gene inserted between *csuA/B* and *csuA* within its *csu* operon. In our colonies of *A. baumannii* 19096T and AB5075, we observed little to no filaments wide enough (72 Å) (Wilharm *et al.*, 2013) to be interpreted as Type IV pili. Only colonies grown at 30 °C and 37 °C displayed a few bacteria with such filaments, while most cells expressed abundant Csu pili. This agrees with the observation that cyclic AMP acts as a master regulator, suppressing the expression of Csu pilus subunits, PNAG-synthesizing enzymes and quorum sensing, and thus biofilm formation in general, while strongly upregulating Type IV pilus subunits and other motility-related genes (Harkova *et al.*, 2024). Therefore, while Type IV pili may contribute to biofilm formation by mediating adhesion to certain surfaces, they likely lose relevance once *A. baumannii* switches to producing Csu pili. However, the expression regulation could be different in other species. Studies of other archaic CUP pili, such as the *P. aeruginosa* CupE system, may help clarify this.”

6. Building off of this, type IV pili appear to not be mentioned at all in the paper despite their presence in this strain of *Acinetobacter baumannii* (I believe). Were any T4P detected during cryo-EM image processing (2D classification steps, .etc)? Given the robust results, I have little doubt that the stacks/networks are formed by the Csu pili, I am just curious whether any T4P were detected at all.

To obtain most of our SPA cryo-EM structures, including the Csu pilus stacks, we expressed Csu pili using our *E. coli* BL21 pBAD-Csu overexpression system. The only exception was the high-resolution structure of the native single Csu pilus, which we purified directly from the *A. baumannii* CCUG 19096T strain through multiple rounds of purification, yielding a highly pure Csu pilus preparation. As a result, our cryo-EM micrographs did not contain Type IV pili, and none were detected during 2D classification or other image processing steps.

7. In figures 2-3 the pili stacks/bundles are quite hard to see in the negative stain TEMs (Fig. 2d-f mostly) as well as in the tomogram slices (Fig. 3a-c). To an untrained eye, which would include much of the broad readership of Nature communications, some of these panels might difficult to interpret even with the helpful annotations. Is there any way that higher contrast images can be used for some of these panels? For figure 2, I have absolutely no idea how to interpret panel 2f. I can detect no difference between stacks and networks in that panel despite the presence of the diamonds. For figure 3 a-c it might be possible to average together several adjacent Z-slices to generate better contrast. Would this be feasible?

We thank the reviewers for this suggestion. We have now revised Fig. 2 and 3 to improve clarity and logical presentation. Please note that the resolution of the TEM figures in the automatically generated pdf file is not optimal. The images are better seen in the docx source files at a 125 % zoom.

We replaced panel 2e with another view that depicts a local network of stacks at a reasonably high resolution. In addition, we added three close-up views (2g–i) of the bundle from panel 2f, which we now call a “superstack”.

We have also revised the figure legend for clarity:

“Fig. 2 | Csu pili form a network of stacks interlinking bacteria in 3D biofilms. a, *A. baumannii* 19096T cells from liquid cultures are mostly covered in separate Csu pili pointing radially in every direction. b–i, A negative stain TEM image of a suspended *A. baumannii* colony reveals a network of Csu pilus stacks (b), including local stacks wrapping around the bacteria (c–e) and a long, fully exposed superstack (f–i) linking two bacteria on the left to a clump of bacteria on the right, which they likely detached from during sample preparation. Typically, the pili growing in the narrow gaps between bacteria wrap closely around each bacterium (c) by forming pairs and flat stacks (d,e) that

further merge into extensive networks connecting multiple bacteria (**b,f**). The ends of the exposed superstack connecting to the two groups of bacteria consist of Csu pilus pairs and stacks emerging from multiple directions (**g,i**) that then merge into a unified stack closer to the centre (**h**). Blue dots highlight examples of separate pili, while red dots border pilus pairs or stacking networks, and blue lines in **d** indicate individual pili in a multi-pilus stack. Diamonds in **e–i** highlight the local networks of pilus stacks merging into the superstack.”

To help confirm that the terms we use above are now clear, we have added a more detailed description of them below. The terms “a network of Csu pilus stacks” and “stacking network” both refer to all Csu pili in a region that are connected to one another through stacking interactions. Due to their length, each individual pilus may take part in multiple separate pairs and stacks, and since the other pili in those pairs and stacks may form even further stacks, this creates an extensive network branching in every direction inside the biofilm. To describe better the large stack connecting the two groups of bacteria, we have now introduced the term “superstack”, which we used to replace the confusing term “bundle” in all of our descriptions of large, mostly organized Csu pilus stacks. The term refers to particularly dense sections of the stacking network that appear in overviews as long, mostly linear filaments. These are quite common, though their low contrast in TEM images and other obscuring factors often prevent us showing them in full. We used the term “superstack” also in our new Supplementary Fig. 1 that depicts stack dimensions observed in our *in vivo* samples.

Finally, to give further examples of Csu pilus stacks in TEM images as well as to answer questions from Reviewer 1, we added a total of four new figures featuring Csu pilus stacks seen in TEM images. First figure depicts the dimensions of the stacks that we have observed *in vivo*, as we mentioned above (Supplementary Fig. 1). Second figure depicts stacks formed in various environmental conditions (pH, excess salt, and different temperatures) (Supplementary Fig. 2). Third figure depicts Csu pilus stacks tethering bacteria in recombinant *E. coli* liquid culture clumps, even when the tip-subunit CsuE has a mutation hindering its ability to bind into hydrophobic surfaces (Supplementary Fig. 4). The fourth one depicts how the Csu pilus stacks link the bacteria together independently of PNAG and eDNA (Fig. 8). These additional figures should make it easier to interpret the contents of Fig. 2.

When it comes to the tomogram Z-slices in Fig. 3, the stacks seen at the tomogram are all at an angle to the plane of the Z-slices, which causes an issue with detecting them, as the reviewers have well pointed out. To make them easier to understand, we averaged 20–25 Z-slices as the reviewers suggested, showing the pili in the stacks slightly better, though unfortunately, the improvement is not much. We also updated the figure legend to indicate, what the tomogram Z-slices now show:

“**a–c**, A representative tomographic slice depicting cell-cell interactions in a fresh, suspended *A. baumannii* colony grown at 22°C (**a**) and close-up views (**b,c**) (red rectangles) with 20–25 Z-slices averaged to show cross-sections of the Csu pilus stacks, as indicated (yellow arrowheads).”

Since averaging several Z-slices had less of an effect than we had hoped, we added an extra video (new Supplementary Video 2) to depict, what the Csu pili and their stacks look like in the tomogram Z-slices. The video tilts back and forth around the view seen in Fig. 3c, while stopping to point out with yellow arrows the Csu pilus stacks that remain in view longer than the individual Csu pili.

Finally, as an additional change to Fig. 3, we added panel 3i, which depicts a stack from the 3D segmentation tilted from 0° frontal view to a 90° side view in four 30° increments. This is similar to what we already had in the current Supplementary Video 1, but we decided it would be good to have it shown in Fig. 3, as well. This should highlight the flatness of the stacks, as from a certain angle they look as narrow as a single pilus.

III. Additional suggestions which may improve the story/results

1. Supplementary Fig. 7 might be better as a main figure as its results are key to the model proposed in current main Figure 7.

We have transferred the former Supplementary Fig. 7 to the Main figures as new Fig. 7, as the reviewers had suggested. Following that, we have added a new Fig. 8 depicting how Csu pilus stacking networks are able to maintain cohesion in bacterial rafts even in the complete absence of PNAG and eDNA.

2. The number of particles used to obtain the type 1 and type 2 pili stacks was quite a lot given the relatively low resolutions achieved. Perhaps heterogeneity is the limiting factor in terms of resolution. I wonder if more classes of pili stacks would be revealed with additional heterogeneity analyses such as cryoSPARC's 3D variability analysis (cluster output might be most useful) or cryoDRGN?

We agree with the reviewers that heterogeneity is a major limiting factor for the resolution of our maps. However, according to our observations, this heterogeneity is mostly from three sources:

1. CryoSPARC misidentifying the near-identical pilus pairs from the type 1 and 2 stacks and combining them into one map by overlapping pilus pairs from one type of stack at variable pair positions in the other stack map (Supplementary Fig. 17e, see also Fig. R3 above). This results in weakened binding junctions in places where the binding junction positions from the other type of stack do not match the correct stack, as well as weak pilus densities poking from the main map. While we sorted out most of the incorrect particles, some of them remain even in the final datasets (see Fig. R6 below).

2. CryoSPARC misidentifying stack orientations and merging 180° flipped particles with the correctly oriented particles (Supplementary Fig. 17d). We recognized this as a similar mixture of flipped views had occurred earlier, when we tried to use SPRING image processing package in our initial attempts to solve the structure of a single Csu pilus rod (Pakharukova *et al.*, 2022). We reduced this error by sorting the stack orientations between 180° flipped search models in Heterogeneous Refinement, while using low O-EM learning rate to prevent the still flipped particles from biasing the maps, before rotating the sorted particles and merging them in Local Refinement (Supplementary Fig. 17b, g, see also our answer to Point 4). However, the issue remains to an extent, resulting in averaged density maps with smeared resolution and weakened binding junctions.

3. Some of the originally picked particles included pairs and individual pili crossing underneath or above the observed stacks (Supplementary Fig. 5). Filament Tracer can avoid these, but it ended up picking far too few particles (leaving out repeating stack regions perpendicular to pilus axes etc.), so we had to use Template Picking instead. Some of these crossing pili remain as background noise.

As the reviewers suggested, we ran cryoSPARC 3D Variability analysis with cluster output for both type 1 and type 2 stack maps. The program revealed no separate clusters that could be identified as novel stack classes (heterogeneity) or structural flexibility, as indicated by the Clustered Scatterplot and the corresponding maps (see **Fig. R6** below). The minor variability detected appears to result from the remaining misidentification of the two types of stacks and stack flipping. In addition, 3D Variability analysis seems to interpret the degrading type 2 stack pilus resolution near the edge of the picked region as variation.

Fig. R6 | 3D Variability analysis of Csu pilus type 1 and 2 stacks using 65,000 particles for each type of stack and cluster output with three clusters. a, b, Clustered scatter plots for type 1 (a) and type 2 (b) stacks. The Cluster 0 is in blue, Cluster 1 in green and Cluster 2 in orange in both plots. Type 1 stack Cluster 0 has 53,380 particles, Cluster 1 has 9,940 particles and Cluster 2 has 1,680 particles, whereas type 2 stack Cluster 0 has 41,255 particles, Cluster 1 has 4,347 particles and Cluster 2 has 19,398 particles. **c, d**, Result Mode

Projections for type 1 (c) and type 2 (d) stacks, highlighting the regions where the analysis detected possible variability. For type 1 stack, the third mode likely correlates with the presence of flipped particles, as indicated by the series of symmetric blobs likely formed by averaging two antiparallel zigzag structures (pilus rods). For type 2 stack, the analysis reveals some flexibility in the orientations of the subunits not involved in the binding junctions, but it also seems to have interpreted the degrading type 2 stack pilus resolution near the edge of the picked region as variation. e–h, End views and side views for type 1 (e) and type 2 (f–h) stack density maps, comparing the Final structure, Cluster 0, Cluster 1 and Cluster 2. Type 2 stack is shown at three different angles (f–h) to make the side views clearer. Cluster map end views indicate that the rod angles remain largely the same as in the final maps. Type 2 stack Cluster 1 map end view shows two extra densities connected to the central pilus pair with 154.29° angles, indicating the presence of a few residual type 1 stack particles overlapped with the central pair.

We do have additional evidence to suggest that there should not be other major stack types apart from the type 1 and 2 stacks. As we stated in our answer to the reviewers' Point 1, our first Homogeneous Refinement of the combined stack data (Supplementary Fig. 17b) generated a map, in which the two stack density patterns are clearly visible at the same time, while overlapping at a 51.43° angle by sharing one Csu pilus pair (Supplementary Fig. 17e, see also Fig. R1b–d above). The type 1 stack end view showed distinct 154.29° angles between three adjacent pili, while the type 2 stack had 102.86° angles, which made it easy to notice that they were different (Supplementary Fig. 17e). Refinement jobs utilizing mixed particle sets repeated this error consistently (Supplementary Fig. 7a, b, see also Fig. R3b above), so this tendency to overlap pairs from different stacks should be very strong, and therefore any unknown stack classes using the same binding junctions should produce similar unexplained density patterns around the central pair. At no stage of the refinement did we observe such unexplained densities. Furthermore, the final stack maps from cryoSPARC Local Refinement (before cropping the maps around the models) connect to additional pilus densities through binding junctions extending in the directions predicted by our stack models, while all other density blobs lack such connections to the stack densities (Supplementary Fig. 17h). This suggests that there should be no additional types of stacks, at least not in significant quantities.

References used in the responses:

- Ahmad I, Karah N, Nadeem A, Wai SN, Uhlin BE (2019) Analysis of colony phase variation switch in *Acinetobacter baumannii* clinical isolates. *PLoS One* 14: e0210082
- Ahmad I, Nadeem A, Mushtaq F, Zlatkov N, Shahzad M, Zavialov AV, Uhlin BE (2023) Csu pili dependent biofilm formation and virulence of *Acinetobacter baumannii*. *NPJ Biofilms Microbiomes* 9: 101
- Anderson SE, Rather PN (2019) Distinguishing Colony Opacity Variants and Measuring Opacity Variation in *Acinetobacter baumannii*. *Methods in molecular biology* 1946: 151-157
- Bentancor LV, Camacho-Peiro A, Bozkurt-Guzel C, Pier GB, Maira-Litran T (2012) Identification of Ata, a multifunctional trimeric autotransporter of *Acinetobacter baumannii*. *J Bacteriol* 194: 3950-3960
- Bohning J, Dobbstein AW, Sulkowski N, Eilers K, von Kugelgen A, Tarafder AK, Peak-Chew SY, Skehel M, Alva V, Filloux A *et al* (2023) Architecture of the biofilm-associated archaic Chaperone-Usher pilus CupE from *Pseudomonas aeruginosa*. *PLoS Pathog* 19: e1011177
- Bohning J, Ghrayeb M, Pedebos C, Abbas DK, Khalid S, Chai L, Bharat TAM (2022) Donor-strand exchange drives assembly of the TasA scaffold in *Bacillus subtilis* biofilms. *Nat Commun* 13: 7082
- Brossard KA, Campagnari AA (2012) The *Acinetobacter baumannii* biofilm-associated protein plays a role in adherence to human epithelial cells. *Infect Immun* 80: 228-233
- Buzzo JR, Devaraj A, Gloag ES, Jurcisek JA, Robledo-Avila F, Kesler T, Wilbanks K, Mashburn-Warren L, Balu S, Wickham J *et al* (2021) Z-form extracellular DNA is a structural component of the bacterial biofilm matrix. *Cell* 184: 5740-5758 e5717
- Cerqueira GM, Peleg AY (2011) Insights into *Acinetobacter baumannii* pathogenicity. *IUBMB Life* 63: 1055-1060

Ching C, Brychcy M, Nguyen B, Muller P, Pearson AR, Downs M, Regan S, Isley B, Fowle W, Chai Y *et al* (2024) RecA levels modulate biofilm development in *Acinetobacter baumannii*. *Mol Microbiol* 121: 196-212

De Gregorio E, Del Franco M, Martinucci M, Roschetto E, Zarrilli R, Di Nocera PP (2015) Biofilm-associated proteins: news from *Acinetobacter*. *BMC Genomics* 16: 933

Gaddy JA, Tomaras AP, Actis LA (2009) The *Acinetobacter baumannii* 19606 OmpA protein plays a role in biofilm formation on abiotic surfaces and in the interaction of this pathogen with eukaryotic cells. *Infect Immun* 77: 3150-3160

Gaines MC, Isupov MN, Sivabalasarma S, Haque RU, McLaren M, Mollat CL, Tripp P, Neuhaus A, Gold VAM, Albers SV *et al* (2022) Electron cryo-microscopy reveals the structure of the archaeal thread filament. *Nat Commun* 13: 7411

Gallagher LA, Ramage E, Weiss EJ, Radey M, Hayden HS, Held KG, Huse HK, Zurawski DV, Brittnacher MJ, Manoil C (2015) Resources for Genetic and Genomic Analysis of Emerging Pathogen *Acinetobacter baumannii*. *J Bacteriol* 197: 2027-2035

Garnett JA, Martínez-Santos VI, Saldaña Z, Pape T, Hawthorne W, Chan J, Simpson PJ, Cota E, Puente JL, Girón JA *et al* (2012) Structural insights into the biogenesis and biofilm formation by the *Escherichia coli* common pilus. *Proceedings of the National Academy of Sciences* 109: 3950-3955

Giles SK, Stroehrer UH, Eijkelkamp BA, Brown MH (2015) Identification of genes essential for pellicle formation in *Acinetobacter baumannii*. *BMC Microbiol* 15: 116

Giraud C, Bernard CS, Calderon V, Yang L, Filloux A, Molin S, Fichant G, Bordi C, de Bentzmann S (2011) The PprA-PprB two-component system activates CupE, the first non-archetypal *Pseudomonas aeruginosa* chaperone-usher pathway system assembling fimbriae. *Environmental microbiology* 13: 666-683

Gohl O, Friedrich A, Hoppert M, Averhoff B (2006) The thin pili of *Acinetobacter* sp. strain BD413 mediate adhesion to biotic and abiotic surfaces. *Appl Environ Microbiol* 72: 1394-1401

Harkova LG, de Dios R, Rubio-Valle A, Perez-Pulido AJ, McCarthy RR (2024) Cyclic AMP is a global virulence regulator governing inter and intrabacterial signalling in *Acinetobacter baumannii*. *PLoS Pathog* 20: e1012529

Nait Chabane Y, Marti S, Rihouey C, Alexandre S, Hardouin J, Lesouhaitier O, Vila J, Kaplan JB, Jouenne T, De E (2014) Characterisation of pellicles formed by *Acinetobacter baumannii* at the air-liquid interface. *PLoS One* 9: e111660

Pakharukova N, Malmi H, Tuittila M, Dahlberg T, Ghosal D, Chang YW, Myint SL, Paavilainen S, Knight SD, Lamminmaki U *et al* (2022) Archaic chaperone-usher pili self-secrete into superelastic zigzag springs. *Nature* 609: 335-340

Pakharukova N, Tuittila M, Paavilainen S, Malmi H, Parilova O, Teneberg S, Knight SD, Zavialov AV (2018) Structural basis for *Acinetobacter baumannii* biofilm formation. *Proc Natl Acad Sci U S A* 115: 5558-5563

Romero M, Mayer C, Heeb S, Wattanavaekin K, Camara M, Otero A, Williams P (2022) Mushroom-shaped structures formed in *Acinetobacter baumannii* biofilms grown in a roller bioreactor are associated with quorum sensing-dependent Csu-pilus assembly. *Environmental microbiology* 24: 4329-4339

Ronish LA, Lillehoj E, Fields JK, Sundberg EJ, Piepenbrink KH (2019) The structure of PilA from *Acinetobacter baumannii* AB5075 suggests a mechanism for functional specialization in *Acinetobacter* type IV pili. *J Biol Chem* 294: 218-230

Rumbo-Feal S, Gomez MJ, Gayoso C, Alvarez-Fraga L, Cabral MP, Aransay AM, Rodriguez-Ezpeleta N, Fullaondo A, Valle J, Tomas M *et al* (2013) Whole transcriptome analysis of *Acinetobacter baumannii* assessed by RNA-sequencing reveals different mRNA expression profiles in biofilm compared to planktonic cells. *PLoS One* 8: e72968

Sonani RR, Liu Y, Xiang J, Cvirkaite-Krupovic V, Du S, Chen X, Krupovic M, Egelman EH (2025) Tat-dependent bundling pilus of a halophilic archaeon assembles by a strand donation mechanism and facilitates biofilm formation. *Proc Natl Acad Sci U S A* 122: e2514980122

Tipton KA, Dimitrova D, Rather PN (2015) Phase-Variable Control of Multiple Phenotypes in *Acinetobacter baumannii* Strain AB5075. *J Bacteriol* 197: 2593-2599

Tomaras AP, Dorsey CW, Edelman RE, Actis LA (2003) Attachment to and biofilm formation on abiotic surfaces by *Acinetobacter baumannii*: involvement of a novel chaperone-usher pili assembly system. *Microbiology* 149: 3473-3484

Tuon FF, Dantas LR, Suss PH, Tasca Ribeiro VS (2022) Pathogenesis of the *Pseudomonas aeruginosa* Biofilm: A Review. *Pathogens* 11

Wang F, Cvirkaite-Krupovic V, Krupovic M, Egelman EH (2022) Archaeal bundling pili of *Pyrobaculum calidifontis* reveal similarities between archaeal and bacterial biofilms. *Proc Natl Acad Sci U S A* 119: e2207037119

Wilharm G, Piesker J, Laue M, Skiebe E (2013) DNA uptake by the nosocomial pathogen *Acinetobacter baumannii* occurs during movement along wet surfaces. *J Bacteriol* 195: 4146-4153

Detailed responses to the reviewers' comments

Second round

Reviewer 1

The authors have successfully implemented all necessary changes and improvements, resulting in a significantly better and more substantial paper. I recommend its publication in its current form.

We are pleased that the reviewer appreciated the revised version of the paper. We sincerely thank the reviewer for the thorough evaluation and for the many valuable suggestions and corrections, which have significantly improved the manuscript.

Reviewers 2 and 3

The authors have more than addressed all of my concerns. I view this work as impactful and within the scope of Nature Communications and I recommend this manuscript for publication. From a cryo-EM perspective the reconstruction of the different classes of pili stacks was technically challenging and impressive, despite the fact that the authors only achieved low to modest resolution of the stacks. Their high-resolution structure of the pilus rod as well as the fact that they use isolated stacks from a Csu *E. coli* expression system leave little ambiguities to the sequence of the pilin subunit in these structures. I have several minor suggestions that I think would improve the manuscript.

We are very pleased to read the reviewer's comment that "the authors have more than addressed all of my concerns." We are grateful for the reviewer's thorough assessment and for the many excellent suggestions and corrections. We have addressed the new minor suggestions as follows:

1. In Figure 5A the authors should include text displaying the resolution of each structure shown.

The resolution is now indicated in parenthesis next to the name of each structure shown in Figure 5. To comply with the editor's request that each figure occupy one page, Figure 5 was divided into Figure 5 (formerly Figure 5a) and Figure 6 (formerly Figure 5b).

2. The authors stated in their rebuttal "To obtain most of our SPA cryo-EM structures, including the Csu pilus stacks, we expressed Csu pili using our *E. coli* BL21 pBAD-Csu overexpression system". It should be made very clear in the methods which expressed systems were used for which structures.

This information has been added to the following sentences in the "Single-particle cryo-EM" section:

"To determine the structure of the native Csu pilus rod, 3.5 µl of sample purified from *A. baumannii* CCUG 19096T strain was applied to glow-discharged 300-mesh gold holey-carbon grids (Quantifoil R1.2/1.3, Electron Microscopy Sciences) and vitrified using a Vitrobot Mark IV (Thermo Fisher Scientific) at 4 °C and 100% humidity."

"To determine the structure of the Csu pilus pair and the initial structure of a Csu pilus stack, 4 µl of sample purified from *E. coli* BL21-AI with pBAD-Csu plasmid was applied to glow-discharged 300-mesh copper holey-carbon grids (Quantifoil R1.2/1.3, Electron Microscopy Sciences) and vitrified in liquid ethane using a Vitrobot Mark IV (Thermo Fisher) at 4 °C and 100% humidity."

"To determine the final Csu pilus stack structures, we used highly concentrated fractions of Csu pili, which we purified from *E. coli* BL21-AI with pBAD-Csu plasmid and then stored at 4°C for over a year."